

# Structure, variability, and origin of the low-latitude nightglow continuum between 300 and 1,800 nm: evidence for $HO_2$ emission in the near-infrared

Stefan Noll[1,2], John M. C. Plane[3], Wuhu Feng[3,4], Konstantinos S. Kalogerakis[5], Wolfgang Kausch[6], Carsten Schmidt[2], Michael Bittner[2,1], and Stefan Kimeswenger[6,7]

[1]Institut für Physik, Universität Augsburg, Augsburg, Germany
[2]Deutsches Fernerkundungsdatenzentrum, Deutsches Zentrum für Luft- und Raumfahrt, Oberpfaffenhofen, Germany
[3]School of Chemistry, University of Leeds, Leeds, UK
[4]National Centre for Atmospheric Science, University of Leeds, Leeds, UK
[5]Center for Geospace Studies, SRI International, Menlo Park, CA, USA
[6]Institut für Astro- und Teilchenphysik, Universität Innsbruck, Innsbruck, Austria
[7]Instituto de Astronomía, Universidad Católica del Norte, Antofagasta, Chile

**Correspondence:** S. Noll (stefan.noll@dlr.de)

**Abstract.** The Earth's mesopause region between about 75 and 105 km is characterised by chemiluminescent emission from various lines of different molecules and atoms. This emission was and is important for the study of the chemistry and dynamics in this altitude region at nighttime. However, our understanding is still very limited with respect to molecular emissions with low intensities and high line densities that are challenging to resolve. Based on 10 years of data from the astronomical

X-shooter echelle spectrograph at Cerro Paranal in Chile, we have characterised in detail this nightglow (pseudo-)continuum in the wavelength range from 300 to 1,800 nm. We studied the spectral features, derived continuum components with similar variability, calculated climatologies, studied the response to solar activity, and even estimated the effective emission heights. The results indicate that the nightglow continuum at Cerro Paranal essentially consists of only two components, which exhibit very different properties. The main structures of these components peak at 595 and 1,510 nm. While the former was previously

identified as the main peak of the FeO 'orange arc' bands, the latter is a new discovery. Laboratory data and theory indicate that this feature and other structures between about 800 and at least 1,800 nm are caused by emission from the low-lying $A''$ and $A'$ states of $HO_2$. In order to test this assumption, we performed runs with the Whole Atmosphere Community Climate Model (WACCM) with modified chemistry and found that the total intensity, layer profile, and variability indeed support this interpretation, where the excited molecules $HO_2$ are mostly produced from the termolecular recombination of H and $O_2$. The

WACCM results for the continuum component that dominates at visual wavelengths show good agreement for FeO from the reaction of Fe and $O_3$. However, the simulated total emission appears to be too low, which would require additional mechanisms where the variability is dominated by $O_3$.



# 1 Introduction

At wavelengths shorter than about 1,800 nm, the Earth's atmospheric radiation at nighttime is essentially caused by non-
thermal chemiluminescence, i.e. photon emission by excited atomic and molecular states that are populated as a result of
chemical reactions. Most of this nightglow emission originates at altitudes between 75 and 105 km in the mesopause region.
The most prominent emitting species are hydroxyl (OH) and molecular oxygen ($O_2$), which cause various ro-vibrational bands
of emission lines from the near-ultraviolet (near-UV) to the near-infrared (near-IR) (Rousselot et al., 2000; Cosby et al., 2006;
Noll et al., 2012). Especially strong emission is found above 1,400 nm, where OH bands of the electronic ground level with a
vibrational level change $\Delta v$ of 2, e.g. OH(3-1), are located. Bands with higher $\Delta v$ that can be found at shorter wavelengths are
significantly weaker. Strong emission is also related to $O_2$(b-X)(0-0) near 762 nm and $O_2$(a-X)(0-0) near 1,270 nm. However,
both bands suffer from strong self-absorption in the lower atmosphere, which makes it particularly challenging to observe any
emission of the former band from the ground. Intrinsically weaker but not self-absorbed $O_2$ bands are (b-X)(0-1) near 865 nm
and $O_2$(a-X)(0-1) near 1,580 nm. Moreover, there are many weak $O_2$ bands at near-UV and blue wavelengths (Slanger and
Copeland, 2003; Cosby et al., 2006). In addition, especially the visual range shows atomic emission lines. Prominent examples
are the atomic oxygen (O) lines at 558, 630, and 636 nm and the sodium (Na) doublet at 589 nm (e.g., Cosby et al., 2006; Noll
et al., 2012).

Apart from individual emission lines, which have a typical width of a few picometres, the nightglow also includes an under-
lying continuum component. It could consist of line emissions if there were such a high line density that even spectroscopic
instruments with high resolving power were not able to distinguish individual lines, i.e. it would be a pseudo-continuum. In any
case, the observation of such a (pseudo-)continuum is more challenging than the study of well-resolved emission lines. The
applied instrument needs to have a sufficiently high resolving power to clearly separate the continuum from the well-known
emission bands and lines. As the continuum can be quite faint compared to the strong lines, even the wings of the line-spread
function and possible straylight inside the instrument can be an issue, together with low signal-to-noise ratios. Moreover, part
of the night-sky radiance is related to extraterrestrial light sources and scattering inside the atmosphere (e.g., Leinert et al.,
1998). In particular, scattered moonlight, integrated and scattered starlight, zodiacal light, and possible light pollution can be
significant sources of radiation. Hence, such components (which might be quite uncertain) need to be subtracted to measure
the nightglow continuum (e.g., Sternberg and Ingham, 1972; Noll et al., 2012; Trinh et al., 2013).

Despite the potential difficulties, Barbier et al. (1951) first noted a possible continuum in the green wavelength range. In
the subsequent decades, additional constraints were found for a continuum in the visual wavelength range between 400 and
720 nm (Davis and Smith, 1965; Broadfoot and Kendall, 1968; Sternberg and Ingham, 1972; Gadsden and Marovich, 1973;
McDade et al., 1986), where the density of strong emission lines is relatively low. This continuum appeared to have a flux of
several Rayleigh per nanometre ($R\,nm^{-1}$) with an increasing trend towards longer wavelengths and a possible local maximum
(or at least plateau) near 600 nm (e.g., Gadsden and Marovich, 1973). Krassovsky (1951) already proposed that this continuum
could be produced by the reaction

$$NO + O \rightarrow NO_2 + h\nu. \tag{R1}$$



The emission produced by this reaction, termed the $NO_2$ air afterglow, was observed in laboratory discharge experiments and has a pressure-dependent maximum, which is located around 580 nm for relevant atmospheric densities (Fontijn et al., 1964; Becker et al., 1972). Space-based measurements of the emission profile showed a peak between 90 to 95 km (von

Savigny et al., 1999; Gattinger et al., 2009, 2010; Semenov et al., 2014a). First indicated by ship-based latitude-dependent measurements (Davis and Smith, 1965) and then studied in more detail with the Optical Spectrograph and Infrared Imaging System (OSIRIS) onboard the Odin satellite (Gattinger et al., 2009, 2010), the emission is about an order of magnitude weaker at low latitudes compared with the polar regions, where typical values near 580 nm are of the order of $10\,R\,nm^{-1}$.

However, Evans et al. (2010) found that an average OSIRIS spectrum for the low latitude range from 0 to $40°\,S$ did not
match the expected spectral distribution of the $NO_2$ air afterglow from Reaction R1 because the data showed a more complex structure with a conspicuous relatively narrow maximum near 600 nm. As an alternative explanation, they proposed emission from electronically excited iron monoxide (FeO) produced by

$$Fe + O_3 \rightarrow FeO^* + O_2, \tag{R2}$$

which had already been identified by Jenniskens et al. (2000) in the persistent train of a Leonid meteor observed by an airborne
optical spectrograph. Their laboratory-based spectrum of these FeO 'orange arc' bands (see also, West and Broida, 1975; Burgard et al., 2006) also matched the OSIRIS spectrum quite well. This interpretation implies that the low-latitude nightglow spectrum around 600 nm can mainly be explained by a pseudo-contiuum consisting of various ro-vibrational bands produced from the FeO electronic transitions $D\,^5\Delta_i$ and $D'\,^5\Delta_i$ to $X\,^5\Delta_i$ (Cheung et al., 1983; Merer, 1989; Barnes et al., 1995; Gattinger et al., 2011a). Based on the small OSIRIS data set covering five 24 h periods, Evans et al. (2010) also found a good correlation
of the pseudo-continuum and the Na chemiluminescence, which also depends on a reaction with ozone ($O_3$) and involves a chemical element supplied by the ablation of cosmic dust (e.g., Plane et al., 2015). Covariations of Fe and Na densities in the mesopause region were previously measured by lidar (e.g., Kane and Gardner, 1993). The corresponding results for the layer heights of both metals also appear to agree well with the results from the OSIRIS data suggesting a 3 km lower continuum emission layer with a peak at about 87 km. The confidence in the FeO scenario further increased by the analysis of nine nights
of sky radiance data obtained from the Echelle Spectrograph and Imager (ESI) at the Keck II telescope on Mauna Kea, Hawaii ($20°\,N$) (Saran et al., 2011). The spectral range from 500 to 680 nm showed a structure with a peak at about 595 nm consistent with laboratory data (West and Broida, 1975). A slight shift of these (and also the OSIRIS) data of about 5 nm towards longer wavelengths could be explained by a higher effective vibrational excitation due to the low frequency of quenching collisions at the lower pressures in the mesopause region (Gattinger et al., 2011a). To date, the most detailed analysis of the shape of the FeO
orange bands and their variability was reported by Unterguggenberger et al. (2017), based on 3,662 spectra of the X-shooter echelle spectrograph (Vernet et al., 2011) of the Very Large Telescope at Cerro Paranal in Chile ($24.6°\,S$, $70.4°\,W$). Clear seasonal variations similar to those of the Na nightglow, which were analysed in the same study, were found. These variations could be characterised by a combination of an annual and a semiannual oscillation (AO and SAO) with relative amplitudes of 17 and 27% and maxima in June/July and April/October, respectively. Strong nocturnal trends were not observed. The
spectrum (after subtraction of other sky radiance components) appeared to have a stable structure. The main peak between 580



and 610 nm with a mean intensity of $23.2 \pm 1.1$ R contributed $3.3 \pm 0.8\%$ to the total emission in the range between 500 and 720 nm.

Unterguggenberger et al. (2017) did not see clear contributions of the reaction

$$Ni + O_3 \rightarrow NiO^* + O_2 \tag{R3}$$

with a bluer spectrum (Burgard et al., 2006; Gattinger et al., 2011b), i.e. with an expected rise of the flux between 450 and 500 nm instead of around 550 nm as in the case of FeO. This is in contrast to the results for an average spectrum of the GLO-1 instrument on the Space Shuttle mission STS 53, where a ratio of the NiO and FeO intensities integrated between 350 and 670 nm of $2.3 \pm 0.2$ was determined (Evans et al., 2011). However, the same study also investigated OSIRIS mean spectra of June/July over a period of three years, which resulted in much smaller ratios of $0.3 \pm 0.1$, $0.1 \pm 0.1$, and $0.05 \pm 0.05$ that better

agree with Unterguggenberger et al. (2017). Evans et al. (2011) also fitted the $NO_2$ contribution from Reaction R1 relative to FeO and found ratios of 0.6, 0.2, and 0.0 with an uncertainty of 0.1. The correlation of these ratios with those for NiO and the extreme variation of the latter suggest large uncertainties with respect to the impact of NiO nightglow.

At wavelengths slightly longer than 700 nm, early publications indicated a significant increase of the radiance (Broadfoot and Kendall, 1968; Sternberg and Ingham, 1972; Gadsden and Marovich, 1973). However, the rocket-based measurement of

McDade et al. (1986) in Scotland (57° N) only showed a moderate radiance of $5.6$ R nm$^{-1}$ at 714 nm and Noxon (1978) measured an average of $7$ R nm$^{-1}$ at 857 nm based on 15 nights at the Fritz Peak Observatory in Colorado (44° N). Low signal-to-noise ratios and the increasing strength of molecular nightglow emission lines (OH and $O_2$) made measurements quite challenging. The latter can also be seen in the shape of the nightglow continuum of the Cerro Paranal sky model (25° S) derived by Noll et al. (2012), based on 874 spectra of the FORS 1 spectrograph covering a maximum wavelength range from

369 to 872 nm. While the region around the FeO main peak (maximum of about $6$ R nm$^{-1}$) looks realistic, the steep rise at the longest wavelengths is obviously related to the low resolving power of only a few hundred.

At wavelengths above 900 nm, Sobolev (1978) provided estimates of about $9$ R nm$^{-1}$ at 927 nm and about $17$ R nm$^{-1}$ at 1,061 nm based on 5 nights of spectroscopic data from Zvenigorod, Russia (57° N). However, a flux of about $16$ R nm$^{-1}$ at 821 nm from the same study is distinctly higher than the result of Noxon (1978) for a similar wavelength. On the other

hand, the Cerro Paranal sky model provides for about $20$ R nm$^{-1}$ at 1,062 nm. In the range between 1,032 and 1,775 nm, the continuum model was coarsely derived from a small sample of 26 near-IR spectra from the relatively new medium-resolution X-shooter spectrograph (Noll et al., 2014), where the quality of the flux calibration and possible instrument-related continuum contaminations were not yet known. In the set of considered wavelengths, the residual continuum (after subtraction of other sky radiance components) shows a minimum (for regions not affected by water vapour absorption) of about $9$ R nm$^{-1}$ at 1,238 nm

and a maximum of about $87$ R nm$^{-1}$ at 1,521 nm. An increased flux level was also measured by Trinh et al. (2013) with the Anglo-Australian Telescope in Australia (31° S) between 1,516 and 1,522 nm. For their sole continuum window, they obtained $30 \pm 6$ R nm$^{-1}$ based on 45 spectra with a resolving power of 2,400, where strong OH lines were suppressed by means of fibre Bragg gratings (Ellis et al., 2012). The data of the covered five nights also indicated a faster decrease of the continuum at the beginning of the night than in the case of the OH lines. Maihara et al. (1993) already measured the range between 1,661



and 1,669 nm with a resolving power of 1,900 in one night at Mauna Kea ($20°$ N) and found $32 \pm 8\,\mathrm{R\,nm^{-1}}$. A similar flux of $36 \pm 11\,\mathrm{R\,nm^{-1}}$ was obtained by Sullivan and Simcoe (2012) between 1,662 and 1,663 nm based on the median of 105 spectra taken with a resolving power of 6,000 at Las Campanas in Chile ($29°$ S). However, the Cerro Paranal sky model provides here only about $13\,\mathrm{R\,nm^{-1}}$. Moreover, 2 h of observations with the GIANO spectrograph at the island La Palma (Spain, $29°$ N) with the very high resolving power of 32,000 (Oliva et al., 2015) revealed a mean continuum level of about $16\,\mathrm{R\,nm^{-1}}$ in the

range from 1,519 to 1,761 nm avoiding regions affected by strong emission lines. Oliva et al. (2015) also estimated that the presence of weak OH emission lines in the window used by Maihara et al. (1993) would require a reduction of the radiance by 65% resulting in about $11\,\mathrm{R\,nm^{-1}}$.

The high uncertainties of the nightglow continuum in the near-IR made it difficult to find explanations for the origin of the emission. The apparent rise of the continuum beyond 700 nm led to the assumption that this could be caused by another

NO-related reaction (Gadsden and Marovich, 1973). As derived by Clough and Thrush (1967) in the laboratory, the reaction

$$NO + O_3 \rightarrow NO_2 + O_2 + h\nu \hspace{6cm} (R4)$$

would be able to produce a broad continuum with a maximum near 1,200 nm. Later, Kenner and Ogryzlo (1984) also investigated the reaction

$$NO + O_3^* \rightarrow NO_2 + O_2 + h\nu \hspace{6cm} (R5)$$

involving excited $O_3$ with an emission maximum near 800 nm. However, the increasing number of continuum measurements did not support a large contribution from these reactions. Finally, calculations by Semenov et al. (2014b) suggested that a radiance maximum of about $15\,\mathrm{R\,nm^{-1}}$ for Reaction R1 would lead to emission maxima of about $5.4\,\mathrm{R\,nm^{-1}}$ for Reaction R4 and about $0.3\,\mathrm{R\,nm^{-1}}$ for Reaction R5, i.e. the reactions of NO with $O_3$ should only be minor contributions in the near-IR especially at low latitudes, where the $NO_2$ air afterglow near 600 nm tends to be much weaker than given by Semenov

et al. (2014b). An alternative proposal for a source of continuum emission was provided by Bates (1993), who suggested metastable oxygen molecules that collide with ambient gas molecules and then form complexes that dissociate by allowed radiative transitions. However, there were no follow-up studies of this scenario. Concerning laboratory measurements, Bass and Benedict (1952) and West and Broida (1975) showed that FeO does not only produce the orange bands. Probably involving different electronic transitions, pseudo-contiuum emission between 400 and 1,400 nm could be measured. It remains uncertain

how strong these additional bands could be under atmospheric conditions.

As there is obviously a lack of knowledge of the structure of the unresolved nightglow emission and its variability (especially beyond the visual range), we studied this topic by means of a large sample of well-calibrated X-shooter spectra similar to those used by Unterguggenberger et al. (2017) for FeO-related research, i.e. mostly in the wavelength range between 560 and 720 nm. For the current study, we considered a much wider wavelength range from about 300 to 1,800 nm. Moreover, the extended data

set covers 10 instead of 3.5 years, which allowed us to perform a more detailed variability analysis. The data processing was also improved (cf. Noll et al., 2022a). We discuss the data set, basic data processing, and extraction of the nightglow (pseudo-)continuum in Sect. 2. In Sect. 3, we then describe the derivation of a mean continuum spectrum, its decomposition





into different components, the seasonal and nocturnal variations of these components, the impact of the solar activity cycle, and an estimate of the effective emission heights. As this analysis revealed that it is necessary to introduce new nightglow
emission processes, we also explored several possible mechanisms for these emissions by carrying out simulations with the Whole Atmosphere Community Climate Model (WACCM) (Sect. 4). Finally, we draw our conclusions in Sect. 5.

## 2  Observations

### 2.1  Data set

The X-shooter spectrograph (Vernet et al., 2011) covers the wide wavelength range between 300 and 2,480 nm with a resolv-
ing power between 3,200 and 18,400 depending on the arm (UVB: 300 to 560 nm, VIS: 550 to 1,020 nm, or NIR: 1,020 to 2,480 nm) and the variable width of the entrance slit with a fixed projected length of 11″. For standard slits with widths of 1.0″ (UVB), 0.9″ (VIS), and 0.9″ (NIR), the current nominal resolving power amounts to about 5,400, 8,900, and 5,600, respectively. The entire X-shooter data archive of the European Southern Observatory from the start in October 2009 until September 2019 (i.e. 10 years of data) was considered for this study. The NIR-arm data have already been used for investigations focusing
on OH emission lines (Noll et al., 2022a, 2023b). As described in these studies, the basic data processing was performed with version v2.6.8 of the official reduction pipeline (Modigliani et al., 2010) and pre-processed calibration data. The resulting two-dimensional (2D) wavelength-calibrated sky spectra were then reduced to one dimension (1D) by averaging along the slit direction and adding possible sky remnants measured in the 2D astronomical object spectrum extracted by the pipeline.

The flux calibration was performed by means of master response curves for different time periods, which we derived from
the comparison of X-shooter-based spectra of spectrophotometric standard stars and the theoretically expected spectral energy distributions (Moehler et al., 2014). As discussed by Noll et al. (2022a), the NIR-arm spectra were calibrated by means of 10 master response curves derived from data of the stars LTT 3218 and EG 274, which have the highest fluxes in that wavelength regime. For the UVB and VIS arms, more data of these stars and additional spectra of Feige 110, LTT 7987, and GD 71 (Moehler et al., 2014) could be used due to the higher flux at shorter wavelengths and the weaker disturbing nightglow emission.
As this increased the sample from 679 to 1,794 spectra and improved the star-dependent time coverage, there were enough data to produce a series of 40 master response curves with a valid period of 3 months on average. This allowed us to better correct the variability of the response, which tends to increase towards shorter wavelengths due to the larger impact of dirt on the mirrors. In the UVB arm at 370 nm, the individual response curves show a relative standard deviation of about 9.1%, whereas this percentage is only about 3.5% at 1,700 nm. From the flux-calibrated standard star spectra, we obtain a residual variability
of 3.6 and 1.7% for the given UVB- and NIR-related wavelengths. Uncertainties of about 2 to 3% are typical for most of the relevant wavelength range. A notable exception are wavelengths around 560 nm, which are especially affected by the dichroic beam splitting (Vernet et al., 2011). There, the flux variations amount to about 4 to 5%. Finally, the absolute fluxes could show wavelength-dependent constant systematic offsets of a few per cent as a comparison of the results for the different standard stars indicate. We removed the differences by taking LTT 3218 as a reference. Hence, the absolute flux calibration depends on
the quality of the theoretical spectral energy distribution of this star (Moehler et al., 2014).



Excluding very short exposures with less than $10\,\mathrm{s}$ and spectra with very wide slits, which are mainly used for the spectrophotometric standard stars, the final sample comprises about 56,000 UVB, 64,000 VIS, and 91,000 NIR spectra. Although the three arms are usually operated in parallel, the numbers differ due to arm-dependent splitting of observations. Failed processing is another, albeit minor, issue. The exposure times can also be different. In general, the sample is highly inhomogeneous due to different instrumental set-ups, a wide range of exposure times up to $150\,\mathrm{min}$, and different possible residuals of the removed astronomical targets. Hence, the selection of a high-quality sample for a specific research goal needs to be done very carefully.

## 2.2 Extraction of nightglow continuum

For the measurement of the OH line intensities in the NIR arm by Noll et al. (2022a, 2023b), lines and underlying continuum were separated by using percentile filters. For the present investigation of the nightglow continuum, we applied the same approach to the other two arms (Fig. 1). As the density and strength of emission lines depends on the wavelength, we used different combinations of percentile and window width in order to optimise the separation. Concerning the percentile, we applied a median filter in the UVB arm, a first quintile filter in the NIR arm, and stepwise transition between both limiting percentiles in the VIS arm. The window width for the major part of the spectral range was 0.008 times the central wavelength (see also Noll et al., 2022a). This width was further modified primarily depending on the line density. In particular, extended relative widths were applied to wavelengths affected by emission bands of $O_2$ (e.g., Noll et al., 2014, 2016) at $865\,\mathrm{nm}$ (0.02 instead of 0.008), $1{,}270\,\mathrm{nm}$ (0.04), and $1{,}580\,\mathrm{nm}$ (0.02). Nevertheless, remnants of these bands could not be fully avoided (see Sect. 3.1).

Compared to the measurement of lines, the continuum separation was performed after two preparatory steps. First, scattered moonlight, zodiacal light, scattered starlight, and thermal emission of the telescope were calculated using the Cerro Paranal sky model (Noll et al., 2012; Jones et al., 2013) and subtracted from the X-shooter spectra (Fig. 1). Note that this is just a rough correction with relatively high systematic uncertainties, especially in the UVB arm when the Moon is up. On the other hand, the sky radiance components related to direct or scattered light of sources from outside the atmosphere are relatively weak in the NIR arm. In particular, around $1{,}500\,\mathrm{nm}$ the nightglow clearly dominates. However, the situation deteriorates beyond $1{,}700\,\mathrm{nm}$, where the non-zero emissivity of the telescope and instrumental optical components leads to a rising thermal continuum depending on the ambient temperature. The second preparatory step was the correction of the atmospheric extinction by scattering and molecular absorption. The former was performed by means of the recipes given by Noll et al. (2012), which consider the change of the reference Rayleigh and Mie scattering from the sky model depending on the wavelength and zenith angle. This correction is mostly relevant for the UVB arm, where flux changes by several per cent are frequent, whereas the effect is negligible in the NIR arm. Note that the nightglow brightness even tends to increase for spectra taken close to the zenith due to Rayleigh scattering. Molecular absorption especially by water vapour but also by $O_3$, $O_2$, $CO_2$, and $CH_4$ reduces the detected radiance. Here, we also used the sky model for a correction. The continuum transmission curve was calculated for the given zenith distance, given amount of precipitable water vapour (PWV), and otherwise standard conditions at Cerro Paranal. For PWV values, we used the results from Noll et al. (2022a) based on intensity ratios of OH lines in the NIR arm


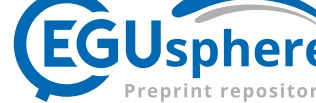

**Figure 1.** Extraction of nightglow continuum for an X-shooter example spectrum with an exposure time of 15 min and standard width of the entrance slit for all three arms, i.e. UVB (a), VIS (b), and NIR (c and d). Wavelengths at the margins of the arm-related spectra and beyond 1,790 nm, which are characterised by continuum data of low quality, are not shown. The original sky spectrum is marked by the cyan (but mostly covered by the red) curve with the green curve as a base line. The latter indicates the modelled contributions by zodiacal light, scattered starlight, and thermal emission of the telescope and instrument. The Moon was below the horizon. Hence, after the subtraction of these components, the cyan spectrum (limited by the dotted zero line) marks the nightglow emission. The red spectrum results from a continuum-optimised correction of the atmospheric extinction, i.e. absorption and scattering. The largest changes compared to the cyan curve are therefore related to wavelength ranges with strong absorption bands. Finally, the black solid curve shows the resulting nightglow continuum based on the application of percentile filters, where the percentile and the width depended on the wavelength range.



with very different absorption fractions. The applied relations were previously calibrated by means of local data from a Low Humidity And Temperature PROfiler (L-HATPRO) microwave radiometer (Kerber et al., 2012). Note that the simple division of a transmission curve does not provide correct results for emission lines as their natural shape is not resolved. However, as we are only interested in the continuum, we can neglect this issue here. As long as the extinction is relatively small, the results of the correction are reasonable. Nevertheless, nearly opaque wavelength regions, e.g. around 1,400 nm due to water vapour

(Fig. 1), cannot be handled in this way. Even if the extinction was exactly known, small uncertainties in the flux calibration and the modelled sky radiance components would make a realistic correction impossible. Hence, the problematic wavelength regions had to be excluded from the analysis.

After the subtraction of the line emission, the continuum spectra were corrected for the increase of the emission with increasing zenith angle due to a longer geometric path through the emission layer. This van Rhijn effect (van Rhijn, 1921) was

calculated assuming that the origin of the extracted continuum was in the mesopause region. The results only weakly depend on the reference height, which we set to 90 km. The validity of the correction is supported by the consistent increase of the continuum flux with increasing zenith angle in the whole wavelength regime for the optimised sample described in Sect. 3.1.

## 3 Results from observations

### 3.1 The mean continuum

For the derivation of the mean nightglow continuum and the variability of the continuum, we only selected the most reliable spectra. As a basic requirement, data products of all three arms with similar temporal coverage had to be available. In the case of arm-dependent differences in the number of exposures (e.g., by shorter exposure times in the NIR arm than in the other arms), the related spectra were averaged, weighted by the exposure time. The most important selection criterion was the minimum exposure time, which was set to 10 min after several tests. The same cut was applied to the VIS-arm sample

studied by Unterguggenberger et al. (2017). This criterion ensures that the signal-to-noise ratio is high. However, the most important effect is the reduction of continuum contamination by bright astronomical sources, which tend to be observed with short exposure times. In order to keep the non-nightglow sky radiance (and the uncertainties of its correction) low, observations with the Moon above the horizon and an illumination of more than 50% were excluded. In the end, these criteria led to 12,723 combined spectra, which constitutes a substantial decrease compared to the full sample. In a second selection procedure,

various features in the continuum probably belonging to the nightglow continuum, residuals of nightglow lines, or residuals of astronomical objects (e.g. the H$\alpha$ line), and the remaining underlying continuum were measured to identify spectra with suspected artefactual contamination (Fig. 2). The resulting selection limits (e.g. non-negative continuum fluxes), which were validated by visual inspection of spectra with values close to the limits, led to a sample of 10,850 spectra. In a third step, the selection was further refined by the search for abrupt changes in the times series of the continuum flux due to the change of the

astronomical target, which suggests a residual contamination. Also validated by visual inspection, this procedure resulted in a final sample of 10,633 combined spectra.





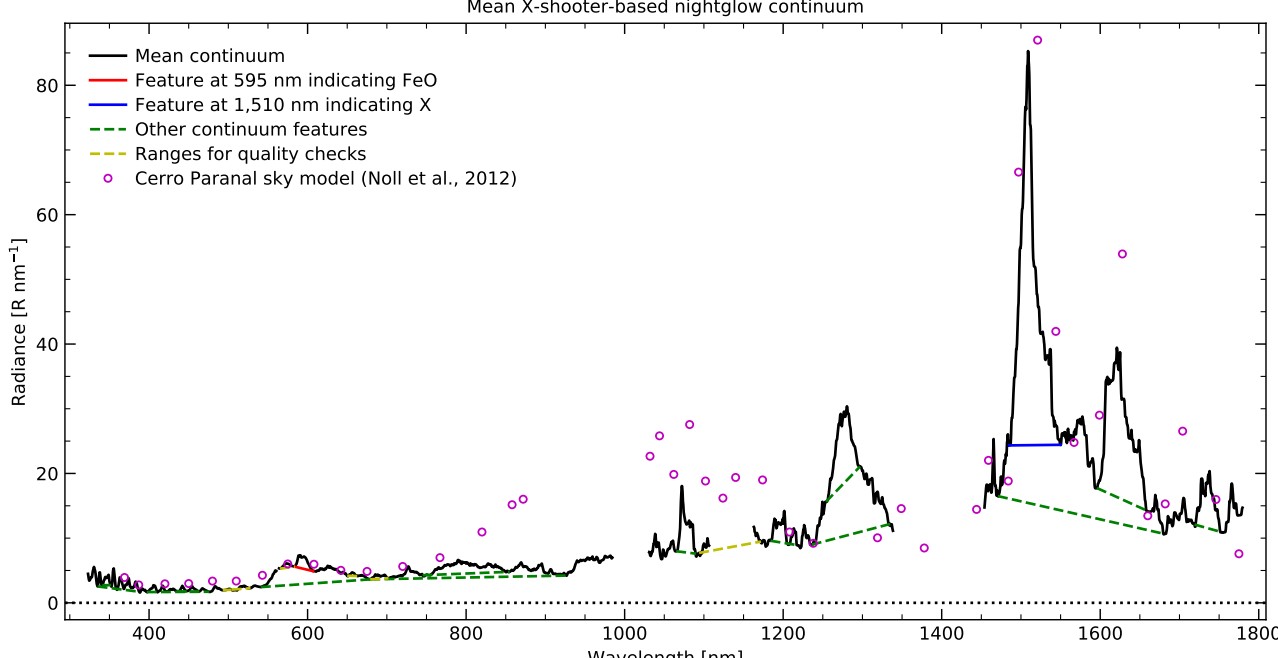

**Figure 2.** Mean nightglow continuum spectrum at Cerro Paranal from 10,633 combined X-shooter spectra. Wavelength ranges with systematic issues were not considered. The plot also shows the wavelength limits for different continuum features and their underlying continua that were used for the sample selection and the scientific analysis. The features centred on 595 nm (red solid line) and 1,510 nm (blue solid line) are the main structures for the latter. Other reliable continuum features (or alternative definitions of their extent) are marked by green dashed lines. The ranges indicated by yellow dashed lines were only used for quality checks (including the detection of the contamination by astronomical objects). They do not mark real nightglow features. For a comparison, the open circles show the mean residual continuum of the Cerro Paranal sky model (Noll et al., 2012).

The mean of this data set is shown in Fig. 2. The spectrum has gaps in wavelength ranges at the margins of the arms (due to high systematic uncertainties) and strong atmospheric absorption (essentially by water vapour). The latter explains the spectral upper limit at 1,780 nm, which also avoids wavelengths with strong thermal emission of the telescope (Fig. 1). At

short wavelengths (i.e. in the UVB arm), various bands related to the electronic upper states c, A′, and A of $O_2$ (e.g., Slanger and Copeland, 2003; Cosby et al., 2006) are visible. As the bands are only partly resolved in the X-shooter spectra, the major part of the emission appears to be present as continuum.

The pronounced step in the continuum at about 555 nm and the peak at about 595 nm indicates the presence of emission from the FeO orange bands (West and Broida, 1975; Jenniskens et al., 2000; Burgard et al., 2006; Evans et al., 2010; Saran

et al., 2011; Gattinger et al., 2011a; Unterguggenberger et al., 2017). The location of the step does not support significant contributions by NiO (Burgard et al., 2006; Evans et al., 2011; Gattinger et al., 2011b), at least from the bluest systems (B-X and C-X), which would already lead to a rise of the flux below 500 nm. The shape of the continuum in this wavelength range



also excludes a significant contribution of $NO_2$ air afterglow (Becker et al., 1972; Gattinger et al., 2009, 2010; Semenov et al., 2014a), which is not unexpected as it is usually only bright at high latitudes (see also Sect. 1). Longwards of the peak at
595 nm, the continuum shows only minor features in the VIS arm with a shallow local maximum at about 800 nm. There, the flux level is not higher than around the FeO main peak and lower than all published continuum measurements in this wavelength range (Sect. 1). At 857 nm, where Noxon (1978) obtained a relatively low value of about $7\,R\,nm^{-1}$, our mean flux is about $5.0\,R\,nm^{-1}$. For a comparison, Fig. 2 also shows the mean continuum from the Cerro Paranal sky model of Noll et al. (2012). While up to 770 nm the model continuum is usually only slightly brighter than our X-shooter-based measurements, the
subsequent three data points are above $10\,R\,nm^{-1}$, which was most probably caused by the use of spectra without sufficient resolving power.

In the NIR arm, our mean continuum is highly structured. In part, these features are related to residuals of blends of strong OH and $O_2$ nightglow emission lines. In particular, remnants of the $O_2$ bands at 1,270 and 1,580 nm related to the transitions (a-X)(0-0) and (a-X)(0-1) can be identified (e.g., Rousselot et al., 2000; Noll et al., 2014, 2016). Nevertheless, these
features only include a very small fraction of the total emissions, which were separated with particularly wide filter windows because of the relatively high line density (see Sect. 2.2). The feature at about 1,080 nm is probably mainly related to the weak $O_2$(a-X)(1-0) band (HITRAN database; Gordon et al., 2022), although the narrow maximum appears to be affected by OH residuals. The most striking continuum feature is certainly the high and narrow peak at about 1,510 nm. It is not related to residuals of strong lines. Hence, it is probably composed of a high number of weak lines, which cannot be resolved with the
spectral resolving power of X-shooter. A feature with a similar origin appears to be the peak at about 1,620 nm.

Both features do not appear to have been discussed previously in the airglow literature. Nevertheless, they are already indicated in the coarse residual continuum component of the Cerro Paranal sky model (Noll et al., 2012), which was also derived from X-shooter spectra (see Sect. 1). Despite the high uncertainties in the model due to premature processing of only a small number of spectra, the majority of the measurement points are relatively close to our mean continuum. Notable exceptions
in the NIR-arm range are the fluxes at 1,628 nm ($54\,R\,nm^{-1}$) and below 1,180 nm. Apart from possible problems with the separation of lines and continuum, the offsets in the latter range suggest systematic issues with the data processing. Data points in ranges that we excluded from our analysis should be treated with caution. In Australia, Trinh et al. (2013) coincidentally performed their continuum measurement of $30 \pm 6\,R\,nm^{-1}$ near the emission peak between 1,516 to 1,522 nm. We find a higher flux of about $50\,R\,nm^{-1}$ for the same range. On the other hand, the mean continuum between 1,661 and 1,669 nm in
Fig. 2 amounts to about $14\,R\,nm^{-1}$, which is clearly lower than the measurements of Maihara et al. (1993) and Sullivan and Simcoe (2012). However, it is slightly brighter than a radiance of about $11\,R\,nm^{-1}$ proposed by Oliva et al. (2015) after the correction of the flux of Maihara et al. (1993) for the contamination by faint OH lines. Compared with the mean continuum flux of about $16\,R\,nm^{-1}$ obtained by Oliva et al. (2015) between 1,519 to 1,761 nm with high resolving power, our corresponding flux of about $22\,R\,nm^{-1}$ is also slightly higher. Apart from differences in the instrumental properties and the data processing,
such discrepancies could also be explained by the different observing sites and observing periods. Oliva et al. (2015) only used 2 h of data taken at La Palma ($29°$ N).



## 3.2 Continuum decomposition

Most of the nightglow continuum emission in Fig. 2 does not exhibit clear features. In order to better understand this emission and its relation to the identified features, we performed a decomposition of the continuum in different components by means of

the wavelength-dependent variability pattern derived from the 10,633 selected spectra. Our approach was to use non-negative matrix factorisation (NMF; e.g., Lee and Seung, 1999), which approximately decomposes an $m \times n$ matrix without negative entries into two matrices with sizes $m \times L$ and $L \times n$ also without negative elements. For this analysis, $m$, $n$, and $L$ are the number of wavelength positions, number of spectra, and number of continuum components, respectively. As we sampled the continuum spectrum with a resolution of $0.5\,\mathrm{nm}$ and only included the ranges indicated in Fig. 2, $m$ was 2,479. For $L$, a

reasonable minimum is 4 since the features correlated with the FeO emission in the VIS arm, the unidentified features in the NIR arm, the $O_2$ features in the UVB arm, and the residuals related to the $O_2$(a-X) bands in the NIR arm should be treated separately. This definition of basic variability classes is supported by a check of the correlations between the variability of the different measured features and continuum windows. In the following, we call these classes FeO(VIS), X(NIR), $O_2$(UVB), and $O_2$(NIR). The names refer to the radiating molecule and location (in terms of the X-shooter arm) of the main features of each

class. It is not excluded that emission of other molecules with a similar variability pattern can contribute. For the application of the NMF, negative fluxes have to be avoided. Because of the thorough sample selection procedure described above, the number of affected data points was very small and negative values could therefore be replaced by zeros without a significant change of the mean spectrum. Only between 1,031 and 1,037 nm (the shortest considered wavelengths in the NIR arm), the mean flux increased by more than 1%. For the derivation of the mean spectrum of each component, we multiplied each of the resulting $L$

component spectra consisting of $m$ data points with the mean of the $n$ corresponding scaling factors.

In the case of an application of the NMF with $L = 4$, it turned out that the $O_2$ component in the UVB arm was not separated from the FeO-related features (similar to $L = 3$). This failure was probably caused by the weakness of the $O_2$ features compared to the other identified continuum structures. As a consequence, we increased the weight of wavelength regions where a crucial feature was relatively strong by the multiplication of suitable factors before the NMF and the division of the same factors in

the resulting component spectra. We tested different numbers and sizes of the windows. In the end, we used 335 to 359 nm, 586 to 603 nm, 1,260 to 1,297 nm, and 1,497 to 1,521 nm, which maximised the weight of the main features of the four variability classes. For finding the best scaling factors, we defined a cost function that uses the relative contributions of the component spectra to the corresponding feature windows as defined above. A simple arithmetic mean of the four fractions favoured solutions with particular large contributions of the two $O_2$-related components. However, the latter can be seen as

contaminations of the FeO(VIS) and X(NIR) components, which are obviously the primary targets of an investigation of the nightglow continuum. Hence, we added the fractions with different weights, finally choosing 0.33 for FeO(VIS) and X(NIR) and 0.17 for $O_2$(UVB) and $O_2$(NIR). Although this procedure is certainly somewhat arbitrary, this choice had relatively little impact on the structure of the solution. It was only important for easily finding a satisfactory solution. Tests showed that the component spectra are relatively similar in wide regions of the parameter space. On the other hand, small changes in the scaling

factors can lead to a very different regime of solutions.



For finding minima of the cost function, we applied a simplicial homology global optimisation (SHGO; Endres et al., 2018) algorithm in the "sobol" mode with 512 sampling points and a limitation of the scaling factors between 1 and 200. The resulting list of local minima for $L = 4$ suggests an uncertainty in the contribution fractions of several per cent for the windows in the UVB and VIS arm and close to 1% for the two windows in the NIR arm. Eventually, we fine-tuned the most promising solution with scaling factors of about 139, 96, 68, and 65 (listed with increasing central wavelength of the feature window) by starting an unconstrained simplex search algorithm (Nelder and Mead, 1965) with the given values as initial parameters. The resulting factors were about 1291, 865, 638, and 597, which differ from the initial values only by a nearly constant factor. This points to a degeneracy of solutions, related to the fact that the values are much higher than 1, i.e. the NMF results appear to be mostly determined by the narrow feature windows. All reasonable local minima found by SHGO in the parameter space are characterised by relatively high values (limited to a maximum of 200), although the ratios of the four factors can clearly differ.

The resulting mean continuum components based on refined simplex search are shown in Fig. 3a. The FeO(VIS) and X(NIR) components contribute to the corresponding feature windows with 83.0% and 95.1%, respectively. Other reasonable solutions tend to show slightly lower percentages. The dominance of these two components extends to wavelengths far away from the main features. While FeO(VIS) dominates almost the entire VIS arm, X(NIR) is the strongest mean component in the NIR arm. Similar contributions appear to be present at the red end of the VIS arm. Below 500 nm, $O_2$(UVB) becomes important with a dominating contribution of 60.5% in the reference range between 335 and 359 nm. Nevertheless, FeO(VIS) appears to still contribute with non-negligible 25.0% there. In terms of the interpretation of this emission as based on FeO, this result is questionable as Reaction R2 should only be exothermic by about 300 kJ mol$^{-1}$ (Helmer and Plane, 1994), which corresponds to a minimum wavelength of about 400 nm. Although the separation of $O_2$(UVB) and FeO(VIS) shortwards of the FeO main peak seems to be the most uncertain result of the NMF-based continuum decomposition, the FeO(VIS) contributions in the UVB arm might support the presence of the blue FeO bands described by West and Broida (1975). With a higher significance, the high contribution of the component at about 800 nm might be explained by the presence of the FeO IR bands (Bass and Benedict, 1952; West and Broida, 1975), although the emission looks smoother than in the laboratory, where it was not produced by Reaction R2. According to the analysis of Gattinger et al. (2011a), the emission of the FeO orange bands is also less structured in the mesopause region than in the laboratory due to a wider distribution of the vibrational populations. Moreover, it is possible that residuals of other emissions in the X-shooter continuum spectra led to an excessive removal of small-scale features. The direct measurement of the broad feature between 745 and 855 nm (Fig. 2) at least shows that the strength of this structure is well correlated with the peak at 595 nm. The measurements in the laboratory found FeO emission up to 1,400 nm. The FeO(VIS) spectrum appears to show a similar extension. However, the uncertainties of the minor contributions in the NIR arm compared to X(NIR) are large.

The FeO(VIS) component could partly be produced by other metal-bearing molecules if their emission showed a similar emission pattern. As already discussed in Sect. 3.1, NiO would be a candidate but the shape of the continuum between 500 and 600 nm does not seem to allow a major contribution. We searched for other possible molecules that could produce a pseudo-continuum in the investigated wavelength regime. A survey of the metal-related chemistry in the mesopause region turned out that another abundant Fe-containing reservoir species (Plane, 2003; Feng et al., 2013; Plane et al., 2015) could be a possible



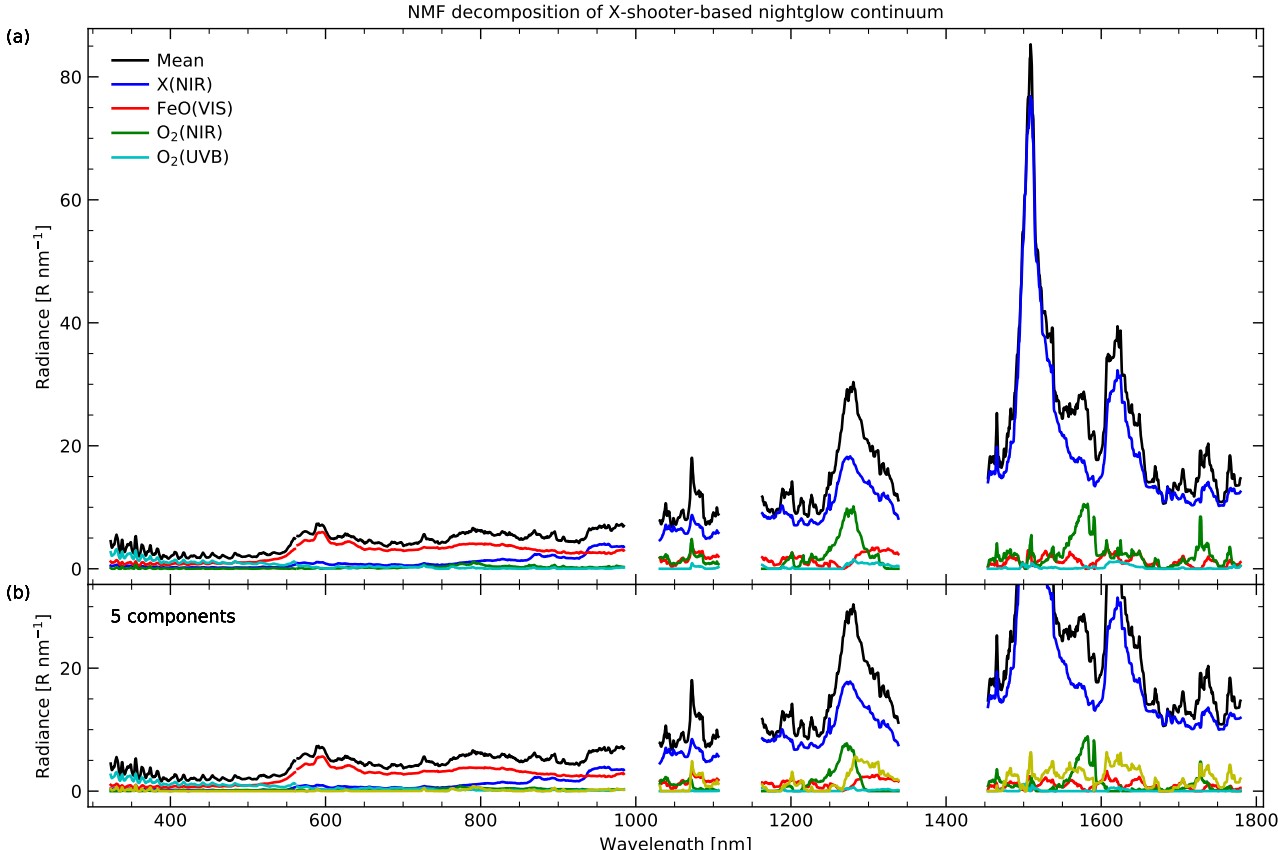

**Figure 3.** Decomposition of mean nightglow continuum spectrum at Cerro Paranal (black curve) into (a) four and (b) five components by non-negative matrix factorisation of the selected 10,633 X-shooter spectra. The details of the procedure are discussed in Sect. 3.2. The four main components are labelled X(NIR), FeO(VIS), $O_2$(NIR), and $O_2$(UVB), which indicates the emitting molecule (if known) and the X-shooter arm with the dominating contribution. The fifth component in (b) appears to mainly consist of residuals of strong nightglow emission lines.

candidate. Unfortunately, chemiluminescence spectra of these molecules do not appear to exist. Nevertheless, inspection of the energetics of the relevant chemical reactions only left the reaction

$$FeOH + O_3 \rightarrow OFeOH^* + O_2 \qquad (R6)$$

as sufficiently exothermic with up to $339\,kJ\,mol^{-1}$ (Sect. 4.1), i.e. almost the entire wavelength range in Fig. 3 could be
covered. We further discuss the possible role of OFeOH emission based on modelling results in Sect. 4.2.

In Fig. 3, the residuals of the strong $O_2$ bands at 1,270 and 1,580 nm are clearly identified by their dedicated component $O_2$(NIR). Nevertheless, the contributions are always smaller than those related to X(NIR). In the range between 1,260 and 1,297 nm, the fraction is only 27.9%. This percentage might be underestimated since X(NIR) shows a similar bump, which





implies that the separation of both components is incomplete. On the other hand, the structure in the mean spectrum is broader
than the $O_2$(a-X)(0-0) band (especially at longer wavelengths), which could suggest that at least a shallow X-related feature is
present in this wavelength range. The weak features at about 1,080 and 1,735 nm are not strong enough to be classified with
the NMF approach.

We checked how the components change if $L$ is set to 5 (keeping everything else untouched). As indicated by Fig. 3b, this
modification appears to mostly affect $O_2$(NIR) by essentially reducing it to the wavelengths of the two strong $O_2$ bands. The
rest is mostly described by the additional component, which seems to be sensitive to any other line residuals (e.g. from OH).
Nevertheless, the version with $L = 4$ is considered as the reference as it is more robust with respect to the important FeO(VIS)
and X(NIR) components, which are slightly weakened in the case of $L = 5$. Tests with even larger numbers of components
only showed a higher complexity without improving the understanding of the nightglow continuum.

**Table 1.** Wavelength positions of $HO_2$ emission bands between 1,000 and 1,800 nm observed in the laboratory in comparison to the
X-shooter-based X(NIR) spectrum

| Upper state[a] | Lower state[a] | Peak[b] (nm) | Band origin[c] (nm) | Presence in X(NIR) |
|---|---|---|---|---|
| $^2A'(002)$ | $^2A''(000)$ | 1,130 | 1,130 | not measured (gap) |
| $^2A'(001)$ | $^2A''(000)$ | 1,270 | 1,257 | moderate strength |
| $^2A'(002)$ | $^2A''(001)$ | (1,290) | 1,280 | possible but blended |
| $^2A'(000)$ | $^2A''(000)$ | 1,430 | 1,423 | not measured (gap) |
| $^2A'(001)$ | $^2A''(001)$ | (1,480) | 1,446 | not clear (partly in gap) |
| $^2A''(200)$ | $^2A''(000)$ | 1,510 | 1,505 | very strong |
| $^2A'(000)$ | $^2A''(001)$ | 1,690 | 1,670 | no clear feature |
| $^2A'(001)$ | $^2A''(002)$ | 1,730 | | weak feature |

[a]  electronic and vibrational ($v_1 v_2 v_3$) levels

[b]  as given by Becker et al. (1974) for low-resolution data (unresolved bands with calculated
wavelengths in parentheses)

[c]  as measured by Becker et al. (1978) and/or Tuckett et al. (1979) at medium/high resolution

If there is only one chemical process that produces the X(NIR) spectrum, the reaction that produces the excited states needs
to be sufficiently exothermic to explain the derived emission at least between about 900 and 1,800 nm. The solution might
be a molecule like OFeOH, where the variability pattern could also be quite different from the FeO emission variations. It is
also possible that the radiating molecule does not include a metal atom if it is sufficiently complex to be suitable to produce
a pseudo-continuum in a wide wavelength range. Here, hydroperoxyl ($HO_2$) appears to be the best candidate. $HO_2$ is often
discussed in terms of mesospheric chemistry with respect to the reaction

$$HO_2 + O \rightarrow OH^* + O_2, \qquad\qquad (R7)$$





which is an alternative production mechanism for vibrationally-excited OH (e.g., Makhlouf et al., 1995; Xu et al., 2012; Panka et al., 2021). The latest results of Panka et al. (2021) suggest that this pathway contributes significantly to the concentration of OH in the lower mesopause region around 80 km, although the resulting vibrational level distribution remains uncertain. The abundance of $HO_2$ in the mesosphere has been observed from the ground (Clancy et al., 1994; Sandor and Clancy, 1998) and

from space (Pickett et al., 2008; Baron et al., 2009; Kreyling et al., 2013; Millán et al., 2015) based on individual lines in the microwave range. While the highest daytime densities tend to be between 75 and 80 km, the weaker nighttime maxima were observed between 80 and 90 km at low latitudes, with the highest altitudes before sunrise (Kreyling et al., 2013). The near-IR spectrum of $HO_2$ has been widely investigated in the laboratory (e.g., Hunziker and Wendt, 1974; Becker et al., 1974, 1978; Tuckett et al., 1979; Holstein et al., 1983; Fink and Ramsay, 1997). Emission was mainly produced by the reaction

$$HO_2 + O_2(a^1\Delta_g) \rightarrow HO_2^* + O_2. \tag{R8}$$

The resulting bands up to 1,800 nm listed by Becker et al. (1974) are given by Table 1. The peak wavelengths are complimented by band origins derived from higher-resolution data of Becker et al. (1978) and Tuckett et al. (1979). In some cases, the provided wavelengths were obtained from the combination of the molecular data of both publications. Most bands in Table 1 are related to transitions between the lowest-lying excited electronic state $^2A'$ and the ground state $^2A''$ that involve the $v_3$ O−OH

stretching vibration of both levels. Interestingly, the excitation energies of $^2A'(001)$ and $O_2(a^1\Delta_g)$ are almost identical. As a consequence, the resulting near-resonant energy transfer produces the $HO_2$ emission feature near 1,270 nm. This is appealing as this would explain our NMF results in this wavelength region. The strongest band in the experiments cannot be checked as wavelengths around 1,430 nm corresponding to the (000-000) band were excluded in our analysis due to the strong absorption by atmospheric water vapour (see Fig. 1). However, the most promising argument for $HO_2$ as X is the only purely vibrational

band in the list. The (200-000) transition that involves the OO−H stretching mode peaks near 1,500 and 1,510 nm (e.g., Hunziker and Wendt, 1974). The second maximum clearly agrees with the peak of our X(NIR) main feature. The invisibility of the first maximum might be caused by systematic uncertainties in the continuum separation near the Q branch of OH(3-1) (Fig. 1) combined with a less pronounced dip at the band origin in the nightglow spectrum.

Other bands of Table 1 that can be checked should peak near 1,690 and 1,730 nm. While we see a possible weak feature in

the X-shooter spectrum in the latter case, there is no clear structure near 1,690 nm. This result is not necessarily an argument against $HO_2$ as the vibronic (000-001) band was much weaker than the (001-000) band near 1,270 nm in the experiment of Fink and Ramsay (1997). A more crucial issue could be the missing evidence for a strong feature near 1,620 nm (Fig. 3) in the laboratory. If $HO_2$ is indeed the correct emitter (i.e. species X), then the population distributions need to be very different in the mesopause region, where the pressure is much lower (3 orders of magnitude) compared to the experiment of Fink and

Ramsay (1997). The spectrum of the latter study that covers the wavelength range between 1,200 and 1,800 nm indicates weaker emission at 1,510 nm than at 1,270 nm. This could point to an increased importance of purely vibrational transitions in the nightglow. Various additional bands might be visible, which could explain the 1,620 nm feature and the relatively high emission over a wide wavelength range. In contrast to X(NIR), the laboratory spectrum does not show significant emission between 1,320 and 1,350 nm as well as below 1,200 nm. The latter is certainly related to Reaction R8, which limits the



emission below 1,270 nm. Nevertheless, Becker et al. (1974) could measure the vibronic (002-000) band near 1,130 nm (in a gap in Fig. 3) and explained it by already vibrationally excited $HO_2$ as reaction partner. In a similar way, Holstein et al. (1983) assumed that two subsequent collisions with $O_2(a^1\Delta_g)$ are required to excite this band and additional weaker bands in the range between 800 and 1,100 nm that involve $^2A'$ $v_3$ states between 3 and 6. The lower wavelength limit for the observed emission would be consistent with the shape of the X(NIR) component.

Importantly, Holstein et al. (1983) found that chemiluminescence can also be generated at wavelengths longer than 800 nm by the main atmospheric production process of $HO_2$ (e.g., Makhlouf et al., 1995)

$$H + O_2 + M \rightarrow HO_2^* + M \tag{R9}$$

with M being an arbitrary collision partner (i.e. $N_2$ and $O_2$ in the mesosphere). Here, the spectrum showed a weaker dependence of the intensities of the vibronic ($00v_3'$-000) bands on $v_3'$ than in the case of collisions with $O_2(a^1\Delta_g)$. The recombination of

H and $O_2$ is also sufficiently exothermic to produce emission potentially as far as about 600 nm. Other chemical reactions producing excited $HO_2$ could also play a role (see Sect. 4). In the view of the remaining uncertainties, we do not replace X by a specific molecule in the following. First, further properties of the unknown emission have to be discussed.

### 3.3 Intensity climatologies

The NMF also returns the scaling factors of each component for each input spectrum. The resulting variability patterns are the

basis for the separation of the components shown in Fig. 3. Before we discuss the variations of the different components, we focus on a comparison of the variability of the two most interesting, directly measured features. These are the peaks at about 595 and 1,510 nm, which are closely related to the NMF components FeO(VIS) and X(NIR). The two peaks were measured by the interpolation between 584 and 607 nm as well as 1,485 and 1,550 nm for the derivation of the underlying continuum (see Fig. 2). The latter feature was then subtracted from the integrated flux in the same wavelength intervals in order to obtain the

feature intensity. Unterguggenberger et al. (2017) already measured the FeO main peak with a similar approach using 3,662 X-shooter VIS-arm spectra taken between October 2009 and March 2013. The continuum spectra were extracted slightly differently by interpolating between wavelengths significantly affected by line emission and leaving the rest of the spectrum untouched. As that method causes noisier spectra than in the case of the percentile filters used in this study (45th percentile and a relative width of 0.008 of the filter at the peak-related wavelengths), the positions for the interpolation on both sides of the

peak were adapted to the corresponding flux minima in each spectrum. Unterguggenberger et al. (2017) reported a reference intensity of the FeO main peak based on a harmonic model of the seasonal variations of $23.2 \pm 1.1$ R. Our sample shows a mean of 27.0 R, which indicates good agreement under consideration of the differences in the sample and the measurement approaches. For comparison, the mean of the peak at 1,510 nm amounts to 1,371 R, i.e. it is about 51 times stronger. The ratio would even be higher for wider feature limits around 1,510 nm that would be reasonable for the X(NIR) component in Fig. 3.

For example, the interval between 1,472 and 1,591 nm would lead to 1,983 R, i.e. a rise by a factor of 1.45 compared to the tighter interval defined in Fig. 2, which we preferred for the measurements in the full spectra in order to avoid the varying contamination by the residuals of the $O_2$(a-X)(0-1) band.





For the study of the variability, we calculated 2D climatologies of local time and day of year in the same way as described in Noll et al. (2023b) for OH emission lines. The measured OH line intensities were not directly used (see also Noll et al., 2022a). Instead, the time series were divided into bins of 30 min and intensities of data with central times in a certain bin were averaged weighted by the exposure time. The reason for this approach was the wide range of exposure times down to 10 s, which could lead to a high weight of a large number of short low-quality exposures (partly clustered in time) in the resulting climatologies if the individual measurements were used. For the NMF-related sample of this study, this is less problematic as only exposures with a minimum length of 10 min were considered. Nevertheless, we also performed this preparatory step for the sake of consistency. Noll et al. (2023b) only selected those bins with a minimum filling of 10 min. This criterion is automatically fulfilled by the NMF-related sample. However, this approach led to a reduction of the number of data points from 10,633 to 7,971 (75.0%).

The climatologies consist of a grid of the centres of the 12 hours between 18:00 and 06:00 LT (the local time related to the solar mean time at Cerro Paranal) and the centres of the 12 months in days of year. The reference values for these grid points were derived from the average of all bins within a radius of 1 h and 1 average month at least if a minimum of 200 bins were selected. In the case of fewer bins, the radius was increased in steps of 0.1 until the criterion was fulfilled. As this issue mainly concerns grid points close to twilight, the temporal resolution at the margins of the climatologies is lower than in the middle of the night. In the LT range between 20:00 and 04:00, the mean relative radius was 1.08. The final climatologies are provided relative to the effective mean, for which the grid point data were averaged weighted by the night contribution (defined by a minimum solar zenith angle of $100°$) of the surrounding cells. Moreover, they are given for a reference solar radio flux at 10.7 cm (Tapping, 2013) averaged for 27 days of 100 solar flux units (sfu). This approach compensates for values between 88 and 110 sfu (with an effective value of 99 sfu) for the different grid points assuming a linear relation between the investigated property and the solar radio flux. The corrections are of the order of a few per cent at most. Hence, the uncertainties in the regression results do not critically affect the quality of the climatologies. The effective intensities of the two features derived from the final climatologies are 27.3 and 1,386 R, which are very close to the mean values for the individual measurements.

In order to better understand the quality of the climatologies, we also calculated them for a minimum sample size of 400 for each grid point as this was the limit used by Noll et al. (2023b) for a total number of bins of up to 19,570. As the NMF-related data set is distinctly smaller, this choice causes smoother climatologies due to the necessary increase of the selection radius. Between 20:00 and 04:00 LT, its mean is 1.43. On the other hand, larger subsamples can reduce the statistical uncertainties. Despite these differences, the intensity climatologies look very similar. The correlation coefficients for the comparison of the versions with lower limits of 200 and 400 bins (only considering grid cells with a nighttime fraction higher than 20%) for the two features are higher than $+0.98$. The impact is larger on the climatologies of the solar cycle effect (SCE), i.e. the relations between the investigated property and the solar radio flux. For this comparison, the coefficients are $+0.86$ and $+0.80$ for the features at 595 and 1,510 nm.

As another test, we investigated the impact of the increase of the total sample size on the climatologies. For the two continuum features, the data selection can be extended as it is only required that they can be measured satisfactorily irrespective of the situation at other wavelengths. As the feature at 1,510 nm is relatively bright, the number of suitable spectra could be increased





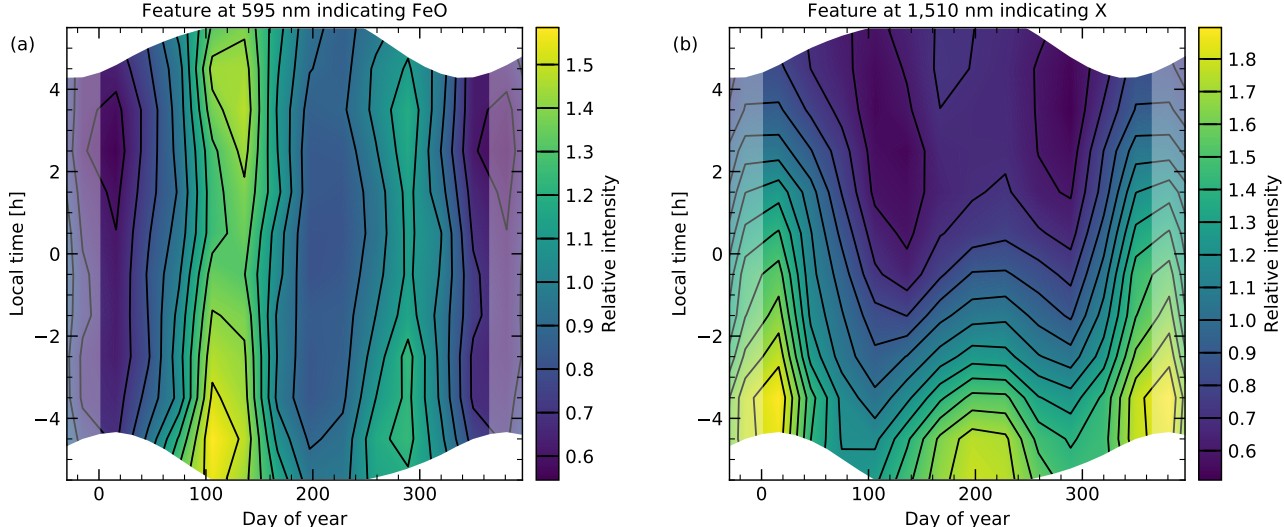

**Figure 4.** Climatologies of intensity relative to the mean as a function of local time (step size of 1 h) and day of year (step size of 1 month) for the continuum features at (a) 595 nm and (b) 1,510 nm based on a sample of 7,971 30 min bins and a minimum subsample size of 200. The climatologies are representative of a solar radio flux of 100 sfu. The coloured contours are limited to times with solar zenith angles larger than 100°. Lighter colours at the left and right margins mark repeated parts of the variability pattern.

to 45,037 including data with minimum exposure times of 3 min (instead of 10 min). This sample resulted in 17,482 30 min bins (an increase by a factor of 2.2), which allowed us to calculate an intensity climatology with a minimum subsample size

of 400 without resolution losses. The result correlates very well with the climatology of the small sample with high resolution. The correlation coefficient $r$ is +0.996. On the other hand, the SCE-related climatology indicates an $r$ of only +0.38, probably partly caused by a vanished outlier in the case of the large sample. Hence, the details of the SCE with respect to LT and day of year remain uncertain, whereas the intensity-related results appear to be quite robust.

In the case of the FeO main peak, the extension of the data set was more limited as the feature is distinctly fainter and

the sample of VIS-arm spectra is smaller. Finally, we selected 22,322 intensity measurements with a minimum exposure time of 5 min, which were converted into 12,785 bins corresponding to an increase of the sample size by a factor of 1.6. The climatology was then also calculated using a minimum subsample size of 400. The resulting intensity variations show a high similarity with those of the small sample as an $r$ of +0.986 indicates. Nevertheless, there appears to be an issue with the large sample with respect to the effective intensity of the climatology, which turned out to be 11.5% higher than in the case of the

small sample. The effective intensity of the 1,510 nm feature only increased by 2.3%. This points to a significant contamination by remnants especially of astronomical objects, suggesting that a relaxation of the selection criteria is problematic for the 595 nm feature. Interestingly, the correlation coefficient for the SCE and the 595 nm feature is +0.76, i.e. it is higher than for the NIR-arm feature. This could be related to a smoother climatology without clear outliers.





As illustrated by the previous discussion, the 2D climatologies of the relative intensity variations of the 595 and 1,510 nm features can be considered as robust. For the NMF-related sample and a minimum subsample size of 200, these climatologies are shown in Fig. 4. The variations of the FeO emission peak in (a) are mainly characterised by a semiannual oscillation (SAO) with maxima in April/May (nightly averaged relative intensity of 1.40) and October (1.17) and minima in January (0.61) and July/August (0.86). The higher intensities for the maxima and minima in April/May and July/August also indicate an annual oscillation (AO) with a maximum in austral autumn/winter. This result is in good agreement with the harmonic fits of the smaller X-shooter data set of Unterguggenberger et al. (2017) (see Sect. 1), which only included spectra until March 2013. WACCM simulations (Feng et al., 2013) suggest that the AO is mainly driven by the Fe concentration, which depends on the meteoric injection rate (maximum in March/April) and subsequent chemical reactions, whereas the SAO is mainly linked to the intra-annual variations of the other FeO-producing reactant, i.e. $O_3$. The concentration maxima of the latter shortly after the equinoxes can also be seen in data of the Sounding of the Atmosphere using Broadband Emission Radiometry (SABER) instrument onboard the Thermosphere Ionosphere Mesosphere Energetics Dynamics (TIMED) satellite (Russell et al., 1999) for Cerro Paranal at 89 km analysed by Noll et al. (2019). Unterguggenberger et al. (2017) also investigated the average nocturnal patterns in the different seasons and found only weak changes without clear trend. With the larger sample of this study, these observations can be confirmed. The changes do not exceed 10 to 20% of the mean value in most parts of the climatology. On average, there is a shallow minimum in the middle of the night. The month-dependent nocturnal variations could be related to the impact of tides. The corresponding features are visible more clearly in the O number density at about 89 km (Noll et al., 2019) and OH emissions especially of lines with relatively high rotational quantum number (Noll et al., 2023b), which are not particularly affected by the rapid nocturnal loss of daytime-produced O close to 80 km.

The 2D climatology of the continuum feature at 1,510 nm in Fig. 4b is very different from the pattern observed for the structure at 595 nm. There is a striking decrease of the intensity by a factor of 2 to 3 from the beginning to the end of the night in all months of the year. Only in the middle of the year in the morning, a plateau appears to be reached. This pattern points to a loss of the excited radiating molecules with the start of the night, i.e. the chemical set-up appears to be different between day and night. Examples of such cases are OH emission especially below 84 km (Marsh et al., 2006; Noll et al., 2023b) due to cessation of $O_2$ photolysis, and $O_2$(a-X) emission (Noll et al., 2016) due to the cessation of $O_3$ photolysis. Interestingly, Trinh et al. (2013) previously reported a decrease of the continuum between 1,516 and 1,522 nm in the first half of the night based on spectra from the Anglo-Australian Telescope (31° S) taken during five nights in September 2011 (see Sect. 1). The decrease in the evening appeared to be slightly faster than in the case of the Q branch of OH(3-1). This is consistent with our results from a comparison with the corresponding OH line climatologies from Noll et al. (2023b), which indicated an about 15% higher intensity reduction between 19:30 and 21:30 LT for the continuum peak on average. The seasonal variations of the 1,510 nm feature show a main maximum in January (nightly averaged relative intensity of 1.59), a secondary maximum in July/August (1.02), and minima in April (0.77) and October (0.84). This behaviour is almost the exact opposite of the seasonal variations of the FeO main peak. The correlation coefficient for the monthly mean values is $-0.90$. This anticorrelation does not seem to support a strong impact of $O_3$ in the production of emitter X. This can be an issue for OFeOH as produced by Reaction R6. On the other hand, the seasonal variability of the 1,510 nm emission is reminiscent of the one expected for atomic hydrogen




**Figure 5.** Climatologies of the scaling factors of the four continuum components X(NIR) (a), FeO(VIS) (b), $O_2$(NIR) (c), and $O_2$(UVB) (d) from non-negative matrix factorisation shown in Fig. 3a. Consistent with Fig. 4, the climatologies are also based on a sample of 7,971 30 min bins and a minimum subsample size of 200.

(H) (Mlynczak et al., 2014), which could be an argument for the participation of H in the production of the radiating molecule.
Interestingly, this is fulfilled by the $HO_2$ production process given in Reaction R9. Based on our WACCM simulations, we discuss this topic in Sect. 4.2 in more detail.

The two discussed features only cover a small part of the corresponding NMF-related component spectra. In the studied wavelength ranges (see Fig. 3), FeO(VIS) and X(NIR) indicate mean intensities of about 2.5 and 9.9 kR (explaining about 18 and 69% of the mean spectrum), i.e. the features at 595 and 1,510 nm have a contribution of about 1.1 and 13.8%. These



percentages further decrease if the radiance in the spectral gaps is roughly approximated by a simple linear interpolation, which results in about 2.9 and 11.8 kR for the two components. In particular, the X(NIR) intensity could further increase as the flux is still relatively high at the upper wavelength limit of 1,780 nm. Apart from the limited wavelength coverage, these values are affected by uncertainties in the separation of the component spectra from other contributions. Nevertheless, the basic structure of the FeO(VIS) and X(NIR) components appears to be realistic. The 2D climatology of the 595 nm feature is well correlated

with the one of the underlying continuum ($r = +0.961$). Moreover, the integrated flux between minima at 679 and 927 nm (Fig. 2) shows a high $r$ of $+0.974$. For the 1,510 nm feature, the $r$ values are even above $+0.99$ if they are calculated for the continuum below the feature or the secondary peak at 1,620 nm measured between 1,596 and 1,662 nm. Even in the continuum below the $O_2(\text{a-X})(\text{1-0})$ band at about 1,080 nm, $r$ is still quite high with $+0.966$. Although partly forced by the wavelength weighting of our NMF procedure, an $r$ of $+1.000$ is remarkable for the correlation of the X(NIR) component (Fig. 5a) and

the 1,510 nm peak (Fig. 4b). The correlation of the FeO(VIS) component and the 595 nm peak is weaker ($+0.926$). The main difference in the 2D climatologies is a lower intensity for the NMF component in the evening compared to the morning (Fig. 5b). This might be caused by the decreasing nocturnal trend in the stronger X(NIR) component, which partly overlaps with FeO(VIS). Thus, the NMF obviously led to more different climatologies than the direct feature measurements showed.

The separation of the two $O_2$-related components probably succeeded due to a relatively weak SAO in the climatologies

(panels (c) and (d) of Fig. 5). The climatological patterns are more reminiscent of the case for O (Noll et al., 2019) with tidal features that are also visible in OH intensity climatologies (Noll et al., 2023b). This similarity is reasonable as the nocturnal production process of these bands is probably related to O recombination (e.g., Slanger and Copeland, 2003) as well as collisions of $O_2$ with excited oxygen atoms in the case of the near-IR emissions (Kalogerakis, 2019). Nevertheless, the correlation coefficient for $O_2(\text{UVB})$ and $O_2(\text{NIR})$ is $-0.22$. The largest discrepancy is present in the evening, when the

intensity of $O_2(\text{NIR})$ steeply decreases due to the decay of the $O_2(\text{a}^1\Delta_\text{g})$ population produced by $O_3$ photolysis at daytime (e.g., Noll et al., 2016), whereas the intensity of $O_2(\text{UVB})$ that is related to electronic states without such a pathway is relatively low. Interestingly, the excess $O_2(\text{a}^1\Delta_\text{g})$ population seems to show an SAO which is consistent with a dependence on the $O_3$ density as in the case of FeO(VIS). The decrease of the intensity with increasing LT might complicate the dynamical separation of $O_2(\text{NIR})$ and X(NIR), which could contribute to the uncertainties around the $O_2(\text{a-X})(\text{0-0})$ band at 1,270 nm in Fig. 3.

### 3.4 Solar cycle effect

As the X-shooter data set covers 10 years between October 2009 and September 2019, the resulting continuum features can also be investigated with respect to the solar cycle. As already discussed in Sect. 3.3, we also calculated 2D climatologies for the SCE. With respect to the features at 595 and 1,510 nm, it turned out that the structures in these climatologies are relatively uncertain. Based on the largest analysed sample for the FeO main peak with 12,785 bins, Fig. 6a indicates the largest positive

SCE values around the austral summer solstice and in the austral winter. The lowest (and possibly negative) values appear to be present around March. Figure 6b for the sample with 17,482 bins of the 1,510 nm feature shows possible maxima in July and November and a minimum in austral autumn, which could possibly be negative.





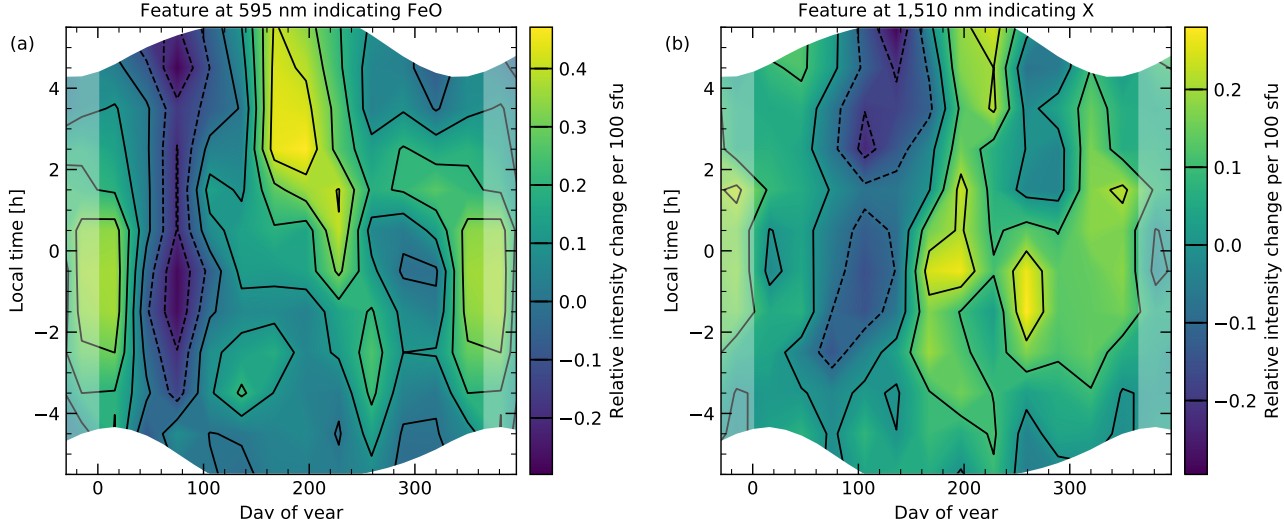

**Figure 6.** Climatologies of the solar cycle effect for the continuum features at (a) 595 nm and (b) 1,510 nm. For each grid point (see caption of Fig. 4), the given value indicates the change of the intensity relative to the corresponding mean for an increase of the solar radio flux averaged for 27 days by 100 sfu. The climatologies were calculated for (a) 12,785 and (b) 17,482 30 min bins and the minimum sample size for each grid point was 400 (cf. Fig. 4).

The resulting effective SCEs derived from the averaging of the 595 and 1,510 nm climatologies are $+10.7$ and $+4.2$ % per 100 sfu, respectively. If these percentages are directly derived from the intensities of the 30 min bins, the results are $+8.1 \pm 1.5$
and $+7.5 \pm 1.4$ % per 100 sfu. For the individual measurements, we obtain $+4.0 \pm 1.2$ and $+6.7 \pm 0.9$ % per 100 sfu. The differences between these results show the uncertainties related to the sample size and the climatological weighting of the data points. In any case, the SCEs for both continuum features are relatively small, which may explain the relatively high uncertainties in the discussed climatological patterns. In contrast, the $O_2$-related features in the continuum show large effects of about $+40$ % per 100 sfu (e.g., using the ranges 335 to 388 nm and 1,254 to 1,297 nm for the NMF-related sample). For
OH, the X-shooter data set indicates line-specific effective SCEs between $+8$ and $+23$ % per 100 sfu (Noll et al., 2023b). On the other hand, chemiluminescent 770 nm potassium (K) emission measured between April 2000 and March 2015 at Cerro Paranal in spectra of the Ultraviolet and Visual Echelle Spectrograph (UVES; Dekker et al., 2000) resulted in a negative effect of $-7.4 \pm 1.3$ % per 100 sfu (Noll et al., 2019). This seems to be related to an even more negative SCE for the K column density, as shown by WACCM simulations for the long period from 1955 to 2005 (Dawkins et al., 2016). For the latitude range
from 0 to $30°$ S, about $-14.4$ % per 100 sfu are given. The same study also provides $-4.7$ % per 100 sfu for the Fe column density. Considering that Fe and K react with $O_3$ to form monoxides that are directly (FeO) or indirectly (KO with subsequent reaction with O) the basis for the chemiluminescence, the difference in the SCEs for the column density of about 10% would support a slightly positive value for FeO nightglow, which would be consistent with our measurements (see also Sect. 4.2).



### 3.5 Effective emission heights

Using the X-shooter NIR-arm data set, Noll et al. (2022a) investigated eight nights in 2017 and seven nights in 2019 with respect to the signatures of passing quasi-two-day waves (Q2DWs) in the intensities of OH emission lines. Q2DWs are only present for a few weeks in austral summer but constitute the strongest wave phenomenon at low southern latitudes (Ern et al., 2013; Gu et al., 2019; Tunbridge et al., 2011). The particularly strong wave between 26 January and 3 February 2017 was used to estimate the effective emission heights of the selected 298 OH lines based on fits of wave phases for a most likely period of

44 h. Apart from the line intensities from the X-shooter data, the study also used OH emission profiles from TIMED/SABER (Russell et al., 1999) for the derivation of the required phase–height relation.

In order to better understand the emission features at 595 and 1,510 nm, which are obviously representative of a large fraction of the nightglow continuum (Sect. 3.3), we also attempted to derive wave phases and the related emission heights for these two features. For a good time coverage during the crucial eight nights, we had to further relax the selection criteria described

in Sect. 3.3. Like the investigated OH lines, the 1,510 nm feature is covered by the NIR arm. With a minimum exposure time of 1 min and only intensities between 200 and 4,800 R, 265 of 388 observations remained in the sample. We manually checked every spectrum and rejected 13 additional spectra with suspicious astronomical targets, i.e. the final sample comprises 252 intensity measurements. Consistent with Noll et al. (2022a), the intensities of 30 min bins were calculated. If the default lower limit of 10 min for the bin filling is used, the resulting sample comprises 82 bins, which is lower than the maximum of

88 for the OH-related sample. However, the sample size can easily be increased to 92 bins if the bin filling threshold is set slightly lower to 8 min. Then, only three bins of the original OH-related sample are lost, all of them present in the evening. As the 595 nm feature is distinctly weaker and the smaller sample of VIS-arm spectra has to be used, the final sample just comprises 125 spectra. The selection criteria include a minimum exposure time of 3 min, the requirement of positive values for the feature and the underlying continuum (for the latter also an upper limit of 18 R nm$^{-1}$), and the rejection of additional

spectra contaminated by the astronomical targets (identification by visual inspection). Then, the binning results in 63 bins if a minimum filling of 8 min is also required (otherwise only 57 bins).

The resulting bin-related intensities normalised by the sample mean were fitted as described in Noll et al. (2022a). The fit formula

$$f(t, t_{\mathrm{LT}}) = c(t_{\mathrm{LT}}) \left( a(t_{\mathrm{LT}}) \cos \left( 2\pi \left( \frac{t}{T} - \phi \right) \right) + 1 \right), \tag{1}$$

contains a cosine with the time $t$ relative to the period $T$ minus a reference phase $\phi$ for 30 January 2017 12:00 LT. The cosine is multiplied by an amplitude $c \cdot a$ and a constant $c$ (which can also be considered as a scaling factor for the mean) is added to this term. As $T$ was set to 44 h, i.e. the optimum derived from the OH data analysed by Noll et al. (2022a), the final fitting parameters were $\phi$, $c$, and, $a$, the latter being the amplitude of the cosine. As the OH time series showed a strong dependence of the amplitude $c \cdot a$ on local time, LT intervals with a length of 1 h centred on $t_{\mathrm{LT}}$ were fitted separately. First, this was done

for the derivation of the optimum phase, which represents the average for the selected LT hours weighted by the inverse of the phase uncertainty. In a second step, the phase $\phi$ was fixed and the LT-dependent parameters $c$ and $a$ were fitted.



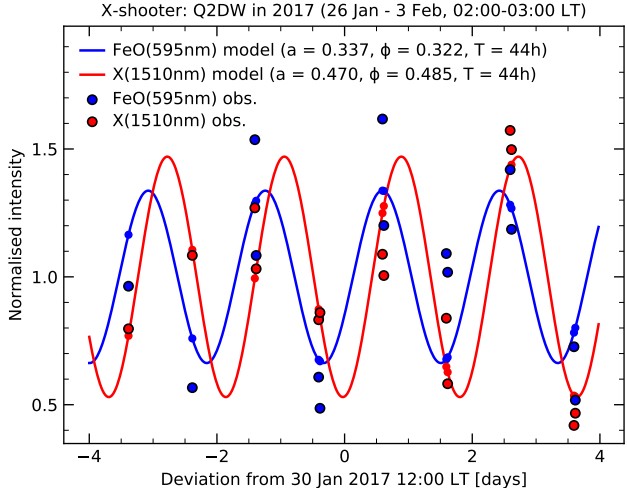

**Figure 7.** Relative intensities of the features at 595 nm (blue) and 1,510 nm (red) for the 14 30 min bins in the interval between 02:00 and 03:00 LT between 26 January and 3 February 2017 and the related fit of a cosine with a period $T$ of 44 h (solid curves with small dots for the effective times of the bins). Measurements and models are given relative to the fitted scaling factor $c$. The remaining fit parameters $a$ (amplitude) and $\phi$ (phase) are provided in the plot.

For the derivation of the phase, only reliable LT intervals with a good time coverage and small phase uncertainties should be used. Noll et al. (2022a) selected four to five LT hours depending on the line. The evening data were always rejected because of low wave amplitudes and a relatively small number of bins. For the 1,510 nm feature, the extended sample with 92 bins

includes the same 39 bins between 01:00 and 04:00 LT that were also considered for the OH-related fits. In the case of the 595 nm feature, the sample with 63 entries shows 38 bins in this range. Alternatively, the fits can be restricted to the interval between 02:00 and 03:00 LT, where all samples include the same (and maximum number of) 14 bins. For the latter case, Fig. 7 shows the measured intensities and the resulting fits divided by $c$ for both continuum features. This normalisation allows one to directly read the amplitude $a$ from the plotted wave fit. The amplitude is higher for the 1,510 nm feature (0.47 vs. 0.34). This

feature also shows a later phase (0.485 vs. 0.322 relative to $T$). Concerning the fit quality, it is clearly visible that the deviations for the strong 1,510 nm feature are distinctly smaller than in the case of the 595 nm feature. The root mean square results in 0.14 compared to 0.23. Nevertheless, the phase for the $\mathrm{FeO}$ main peak does not seem to be less robust since the standard deviation for the independent fits of the three intervals between 01:00 and 04:00 LT indicates 0.033 compared to 0.042 for the peak at 1,510 nm. The mean phase from these intervals is slightly higher in both cases (0.489 and 0.339).

For both continuum features, Fig. 8 shows the LT-dependent amplitudes $c{\cdot}a$ and scaling constants $c$ for an optimum phase $\phi$ that is only based on the interval centred on 02:30 LT. For 595 nm in (a), there were only sufficient data (at least seven bins) for a fit in the LT range between 23:00 and 04:00 LT. Hence, the situation in the evening remains unclear. For the covered time range, the amplitude relative to the mean is about 0.2 with a peak of 0.32 for 02:00 to 03:00 LT, i.e. the decisive interval for the phase derivation. The constant $c$ is around 1, which indicates that there was not a clear trend of the mean with local time.



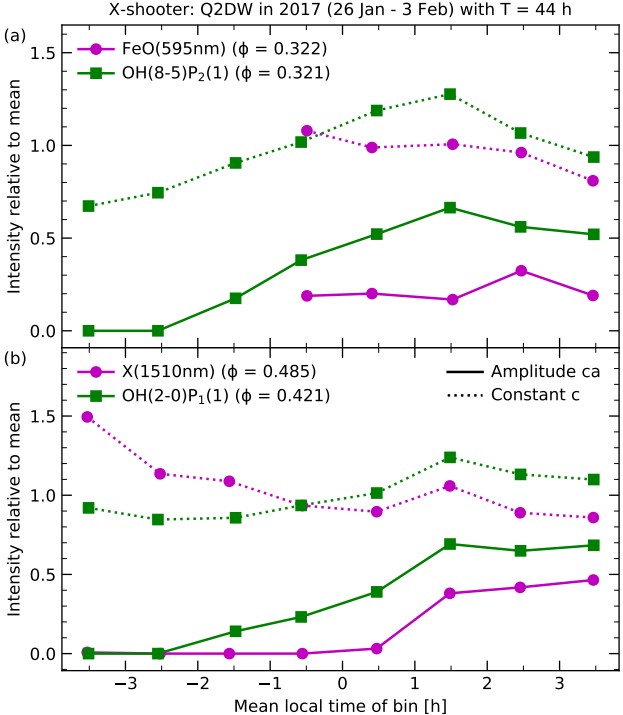

**Figure 8.** Amplitudes $c \cdot a$ (magenta solid curves with circles) and scaling constants $c$ (magenta dotted curves with circles) relative to the sample mean as a function of local time (step size of 1 h; only intervals with sufficient data) for cosine fits of the Q2DW in 2017 with a period of 44 h based on intensity data of the (a) 595 nm (sample of 63 bins) and (b) 1,510 nm (sample of 92 bins) features. The given phases are only based on fits of the interval between 02:00 and 03:00 LT. In addition, the corresponding curves for OH emission lines (green curves with squares) with similar phases $\phi$ at 30 January 12:00 LT from the same fitting procedure are shown (cf. Noll et al., 2022a).

We compare these curves with those of an OH line with almost the same phase for the 02:30 LT interval. Note that the given $\phi$ of 0.321 is slightly lower than the value of 0.328 in the data release of Noll et al. (2022b), which is based on several LT hours. The data for OH(8-5)P$_2$(1) indicate a significantly larger maximum amplitude $c \cdot a$ of 0.71 (between 01:00 and 02:00 LT). Even if the different scaling factors are considered and only the cosine amplitudes $a$ are compared (0.54 vs. 0.34 between 02:00 and 03:00 LT), the impact of the Q2DW on OH lines appears to be stronger, which probably reveals dynamical differences if O$_3$

reactions with Fe and H are compared. The remarkable decrease of the wave amplitude towards the beginning of the night is not covered by the data for the 595 nm feature. Hence, a clear difference in the nocturnal trends in the joint LT range is not obvious.

     The 1,510 nm data in Fig. 8b show the entire OH-related night interval. The scaling factor $c$ indicates a clear decrease in the course of the night, which implies that the average nocturnal behaviour as shown in the climatology in Fig. 4b is also relevant

for the eight nights affected by the strong Q2DW. On the other hand, there is no detection of this wave except for the last three intervals between 01:00 and 04:00 LT, which explains why only this range was useful for phase fits. There $c \cdot a$ is clearly larger





than for the FeO main peak (like $a$ in Fig. 7). We compare the 1,510 nm feature with OH(2-0)P$_1$(1), which shows the highest phase for the studied OH lines. The nocturnal trend in $c \cdot a$ of this line seems to be roughly consistent, although the increase in the middle of the night starts earlier and is slower. These differences might be related to the remaining $\Delta\phi$ of 0.064. The

discrepancies between the absolute $c \cdot a$ of continuum feature and OH line appear to be mainly related to the different nocturnal mean behaviour. The deviation of the 1,510 nm feature with respect to the cosine amplitude $a$ is relatively small at the end of the night (0.54 vs. 0.62 for 03:00 to 04:00 LT), i.e. the responses to the passing Q2DW appear to be comparable. The agreement in the nocturnal development of the amplitude also suggests that the high phase value for the 1,510 nm feature is realistic.

As described by Noll et al. (2022a), the X-shooter-based reference phases of the Q2DW were converted into heights by

using the linear phase–height relation derived from altitude-dependent wave fits of 22 OH emission profiles in the SABER 2.1 μm channel (Russell et al., 1999), which were taken around Cerro Paranal in the eight relevant nights at about 04:00 LT. The regression for the height range from 80 to 97 km resulted in an intercept of $3.027 \pm 0.049$ at 0 km and a vertical wavelength of $31.74 \pm 0.56$ km. Noll et al. (2022a) also applied a correction that considers differences in the properties of the X-shooter and SABER samples. Based on a comparison of phase fits for the vertically integrated emission profiles for both OH-related

SABER channels and the phases from the X-shooter data weighted for the transmission curves of these channels, a general shift of the heights by $-0.43 \pm 0.13$ km was performed. As the phases slightly change if only the LT interval between 02:00 and 03:00 is used for their derivation as described above, we recalculated this shift and found about $-0.79$ km. As a result (which also considers the systematic decrease in $\phi$), the mean effective emission height of all 298 OH lines was 0.10 km lower than given by Noll et al. (2022a). If the mean of the phases from the three intervals between 01:00 and 04:00 LT

is used instead, the resulting offset of $-0.45$ km is very close to the original value and the mean height decreases just by 0.05 km. Hence, the change in the calculation of the optimum phase does not appear to have a significant effect on the resulting emission heights. Using the SABER-based phase–height relation and X-shooter-based phases for the 02:30 LT interval with the estimated corrections (i.e. $-0.79$ km and $+0.1$ km to be consistent with the published OH-related heights), we finally obtain altitudes of 85.2 and 80.0 km for the 595 and 1,510 nm features, respectively. From the comparison of heights for OH lines

with the same or similar ro-vibrational upper levels, Noll et al. (2022a) found uncertainties of several tenths of a kilometre. If we take the reported phase standard deviations of about 0.04 as derived from the LT hours between 01:00 and 04:00 for the two continuum features as an indicator, then the uncertainties might even be of the order of 1 km. Moreover, the use of the full time series is important for the quality of the results. For example, the heights would be unrealistically high (near 100 km) if the first night was excluded from the fits. However, the difference between the values for both features would only slightly

change and rejecting the last night would only have a minor effect.

The given altitudes are representative of the effective height for the strongest absolute variations related to the passing Q2DW. They differ from the effective mean height of the emission. Noll et al. (2022a) found that the average centroid emission altitude for the two OH-related SABER channels at Cerro Paranal (Noll et al., 2017) was about 4.07 km higher than in the case of the corresponding variability-related heights. Significant discrepancies are not surprising in this context since the steep

decrease of the O number density in the lower parts of the OH emission profile (e.g., Smith et al., 2010) lead to stronger relative intensity variations towards lower heights. Nevertheless, the amount of the discrepancy is quite high, which might be explained





by the large amplitude of the Q2DW. It is questionable whether the effective emission heights for the continuum features need to be shifted by a similar value. However, as OH and FeO are produced by reactions that involve $O_3$, it is not unlikely that the impact of the O profile is similar for the 595 nm peak at least. Moreover, the variability-related height for FeO is well in 700 the range between 81.8 and 89.7 km found for the OH lines by Noll et al. (2022a). Thus, also assuming a shift of 4.07 km, we would obtain a centroid altitude of 89.2 km. This value appears to be close to other observations. For the FeO orange bands, Evans et al. (2010) measured with OSIRIS on Odin between 0 and 40° S in April/May 2003 a centroid emission height slightly (up to 1 km) higher than the peak at about 87 km. Modelling of the FeO layer involving FeMOD (Gardner et al., 2005) at 20° N in March 2000 even resulted in a peak height of 89.5 km (Saran et al., 2011).

Without a good knowledge of the chemistry related to the 1,510 nm feature, the difference between mean centroid and Q2DW-related effective emission height is uncertain. If the OH-based shift is applied, the former would be about 84.1 km. This is possibly an upper limit and indicates that the emission layer appears to be lower than the OH and FeO layers. Previous simulations of the Fe-related layers with WACCM by Feng et al. (2013) showed that the densities of neutral molecular reservoir species such as $FeO_3$, $FeOH$, and $Fe(OH)_2$ can peak several kilometres lower than the FeO density. Modelling also 710 suggests that the $HO_2$ density maximises near 80 km (Makhlouf et al., 1995). Such altitudes were also obtained from nocturnal microwave measurements of $HO_2$, although the density peaks could also be several kilometres higher in the second half of the night (Kreyling et al., 2013; Millán et al., 2015). Overall, the discussed candidates for emitter X appear to produce emission at heights consistent with our measurements. With our optimised WACCM simulations, we can discuss the height distributions more quantitatively (see Sect. 4.2).

**4   Modelling**

**4.1   Model set-up**

For a better understanding of the X-shooter-based nightglow continuum and its variability as discussed in Sect. 3, we performed dedicated WACCM simulations. Community Earth System Model (CESM1, WACCM4) simulations with metal chemistry have previously been used for combined observational and modelling studies of chemiluminescent FeO (Unterguggenberger et al., 720 2017) and $K(4^2P)$ emissions (Noll et al., 2019) above Cerro Paranal. Here, we carried out modelling simulations from the updated version of CESM2 (WACCM6) with Na and Fe chemistry to check the FeO-related results and to explore potential candidates for the new pseudo-continuum. CESM2 (WACCM6) is described by Gettelman et al. (2019). Na and Fe chemistry is updated based on Plane et al. (2015). The meteoric injection function (MIF) of Fe is from Carrillo-Sánchez et al. (2016), which is different to that used in Feng et al. (2013). We divided the Fe MIF by 5 to match lidar observations (e.g., Daly et al., 725 2020). Here, we use the specified dynamics version of WACCM6 nudged with NASA's Modern Era Retrospective Analysis for Research and Application MERRA2 reanalysis data set (Molod et al., 2015). The model has a resolution of 1.9° in latitude and 2.5° in longitude and contains 88 vertical levels from the surface to 140 km. The simulation covers the period from 1 Jan 2003 to 28 Dec 2014 (Universal Time). Monthly mean values of selected variables were calculated to save disc space. The model





output was also sampled every half an hour for 24° S and 70° W near Cerro Paranal and interpolated in the height range from
40 to 130 km with a step size of 1 km. The latter is used for the analysis in Sect. 4.2.

**Table 2.** Emission from electronically excited Fe-containing molecules

| Number[a] | Reaction | Rate coefficient[b] $(cm^3 molecule^{-1} s^{-1})$ | Reference |
|---|---|---|---|
| R2 | $Fe + O_3 \rightarrow FeO^* + O_2$ | $2.9 \times 10^{-10} e^{-174/T}$ | Feng et al. (2013) |
| R6 | $FeOH + O_3 \rightarrow OFeOH^* + O_2$ | $7.3 \times 10^{-10}(-200/T)^{-1.65}$ | This work |
| R10 | $FeO + O_3 \rightarrow FeO_2^* + O_2$ | $3.0 \times 10^{-10} e^{-177/T}$ | Rollason and Plane (2000) |
| R11 | $OFeOH + O \rightarrow FeOH + O_2$ | $6.0 \times 10^{-10}(-200/T)^{-1.68}$ | This work |
| R12 | $OFeOH + FeOH \rightarrow MSP^c$ | $9.0 \times 10^{-10}$ | This work |
| R13 | $OFeOH + OFeOH \rightarrow MSP^c$ | $9.0 \times 10^{-10}$ | This work |

[a]   consistent with numbering in text

[b]   temperature $T$ in Kelvin

[c]   meteoric smoke particles

**Table 3.** Potential mechanisms for generating $HO_2$ emission

| Number[a] | Reaction | Rate coefficient $(cm^3 molecule^{-1} s^{-1})$ | Reference |
|---|---|---|---|
| R8 | $HO_2 + O_2(a^1\Delta_g) \rightarrow HO_2^* + O_2$ | $1.0 \times 10^{-10}$ | This work |
| R9 | $H + O_2 + M \rightarrow HO_2^* + M$ | $k_0(4.4 \times 10^{-32}, n = 1.3),$ $k_\infty(7.5 \times 10^{-11}, m = -0.2)^b$ | Burkholder et al. (2015) |
| R14 | $H + O_3 \rightarrow HO_2^* + O$ | $7.0 \times 10^{-13}$ (upper limit) | Howard and Finlayson-Pitts (1980) |
| R15 | $HO_2 + O \rightarrow OH + O_2(a^1\Delta_g)$ | $0.95 \times 2.7 \times 10^{-11} e^{222.5/T}$ | This work[c] |
| R16 | $HO_2 + O \rightarrow OH + O_2(X^3\Sigma_g^-)$ | $0.05 \times 2.7 \times 10^{-11} e^{222.5/T}$ | This work[c] |

[a]   consistent with numbering in text

[b]   low- and high-pressure limits with exponents $n$ and $m$ for $(T_{ref}/T)$ with $T_{ref} = 300$ K (three-body recombination)

[c]   c.f. Burkholder et al. (2015)

Table 2 lists potential Fe-related nightglow chemistry that was explored. Reaction R2 generates FeO emission (Feng et al.,
2013). According to Helmer and Plane (1994), it is exothermic by $301 \pm 8$ kJ mol$^{-1}$, i.e. almost the entire wavelength range
of the X-shooter nightglow spectrum could be covered. Reaction R10 that produces $FeO_2$ indicates a similar exothermicity of
$311 \pm 48$ kJ mol$^{-1}$ (Rollason and Plane, 2000). For the other reactions in Table 2, the reaction exothermicities were calculated
using the high accuracy complete basis set CBS-QB3 method (Frisch et al., 2016). The production of OFeOH via Reaction R6





is sufficiently exothermic by $339\,\mathrm{kJ\,mol^{-1}}$ if $\mathrm{O_2(X^3\Sigma_g^-)}$ is produced, or $244\,\mathrm{kJ\,mol^{-1}}$ if $\mathrm{O_2(a^1\Delta_g)}$ is the product (which is the spin-conserving channel). OFeOH has an abundance of low-lying electronic states (eight states below $2.5\,\mathrm{eV}$) which could be involved in emission from the visual to the near-IR. Reaction R11 is exothermic by $60\,\mathrm{kJ\,mol^{-1}}$. Both reactions should not have significant barriers and hence calculated capture rate coefficients (Georgievskii and Klippenstein, 2005) are assigned
to these reactions. Reactions R12 and R13 represent polymerisation of OFeOH and FeOH to make meteoric smoke particles, which are treated as a permanent sink.

The first three reactions in Table 3 constitute potential mechanisms for the generation of excited $\mathrm{HO_2}$, an attractive candidate for the source of the X(NIR) component of the nightglow continuum measured by X-shooter. We list the already mentioned Reaction R8 that is most relevant for the production of chemiluminescent emission in the laboratory and Reaction R9 that
also produces chemiluminescence and is the main production process of $\mathrm{HO_2}$ (Sect. 3.2). In principle, the direct radiative recombination of H and $\mathrm{O_2}$ could also contribute. However, this mechanism is very unlikely to compete with the termolecular Reaction R9 at the relatively high pressures of the mesopause region. It would need a probability of the order of $10^{-8}$ to make a small contribution at least. As confirmed by tests, the intensity variations from this reaction are very similar to those of the termolecular case. Hence, an estimate of the relevance would be very challenging. We neglect this possible minor channel
in the following. Lastly, the reaction between H and $\mathrm{O_3}$ (Reaction R14) is sufficiently exothermic to produce excited $\mathrm{HO_2}$. However, the channel producing $\mathrm{HO_2 + O}$ is known from experiment to be minor (3%) compared with the channel producing $\mathrm{OH + O_2}$ (Howard and Finlayson-Pitts, 1980).

Reactions R15 and R16 lead to the destruction of $\mathrm{HO_2}$ by collisions with O. The difference between both reactions is the electronic state of the resulting $\mathrm{O_2}$ molecule. We included this distinction as we consider Reaction R15 as the source of the
nighttime production of $\mathrm{a^1\Delta_g}$. With a branching ratio of 95%, we explore the maximum possible contribution of this pathway. For a better understanding of the impact of this percentage, we also performed a simulation with a relative $\mathrm{a^1\Delta_g}$ yield of 40%. The main effect is the decrease of the nocturnal $\mathrm{HO_2}$ emission due to Reaction R8 around midnight and later in agreement with the reduced percentage. At the beginning of the night, the impact is smaller as $\mathrm{a^1\Delta_g}$ mainly originates from the daytime $\mathrm{O_3}$ photolysis, which is considered in WACCM. The decay of this population shows a time constant of the order of 1 hour in
the mesopause region (Noll et al., 2016). There are other possible mechanisms that could contribute to the nighttime $\mathrm{a^1\Delta_g}$ population. However, there is a remarkable lack of knowledge with respect to the efficiency of these reactions. The 'classical' pathway via O recombination and subsequent collisions (e.g., Barth and Hildebrandt, 1961; Slanger and Copeland, 2003) that is important for the production of the $\mathrm{b^1\Sigma_g^+}$ and higher electronic states does not appear to lead to a sufficient $\mathrm{a^1\Delta_g}$ population, including heights around the peak of the $\mathrm{O_2(b\text{-}X)(0\text{-}0)}$ band at $762\,\mathrm{nm}$ near $94\,\mathrm{km}$ (e.g., Yee et al., 1997). $\mathrm{O_2(a\text{-}X)(0\text{-}0)}$ at
$1{,}270\,\mathrm{nm}$ shows a mean centroid emission height of about $89\,\mathrm{km}$ at Cerro Paranal (Noll et al., 2016). This altitude is similar to the centroid emission heights of OH lines, particularly those with relatively high vibrational excitation (Noll et al., 2022a). Therefore, reactions between OH and O might lead to the generation of excited $\mathrm{O_2}$ molecules in the altitude range of the OH emission. Another source of $\mathrm{a^1\Delta_g}$ at even lower heights could be Reaction R9 if M is $\mathrm{O_2}$. However, the efficiency is quite uncertain. Hence, we focus on Reaction 16, which also provides $\mathrm{a^1\Delta_g}$ at heights relevant for $\mathrm{HO_2}$. We evaluate this choice in
Sect. 4.2.



To be consistent with the analysis of the nocturnal/seasonal variations in the X-shooter-based measurements (Sect. 3.3), we derived model climatologies for the vertically integrated volume emission rates between 40 and 130 km in a similar way. We only considered nighttime data with solar zenith angles larger than $100°$, which reduced the sample from 210,240 to 92,064 time steps for 12 years. The time resolution of 30 min is consistent with the binning of the X-shooter data. As the resulting

sample is still much larger than those used for the climatologies of the binned measured data, it is was not necessary to partly decrease the resolution to achieve minimum subsample sizes of 200 or 400 for each relevant grid point (Sect. 3.3). Consistent with the X-shooter-related results, the intensity climatologies are given for a fixed solar radio flux averaged for 27 days of 100 sfu. Although the covered time interval from 2003 to 2014 only partly overlaps with the X-shooter sample (Oct 2009 to Sep 2019), the mean solar radio flux of all nighttime climatological grid points was also very close to this reference value (97 to

107 sfu with a mean of 100 sfu), which caused only very small corrections. We also compared the effective solar cycle effects for the nighttime climatologies of the measured and modelled emissions. The shift of the time interval certainly contributes to the systematic uncertainties but does not appear to significantly increase them. Based on the results for the different calculation methods discussed in Sect. 3.4, we expect total uncertainties of the order of a few per cent per 100 sfu. Apart from intensity climatologies, we also derived climatologies for the centroid heights of the emissions, i.e. emission-weighted heights for the

whole range from 40 to 130 km. The typical nighttime emission profiles were mostly well localised in the mesopause region. Finally, we calculated climatologies for the number densities of different relevant atmospheric species at specific heights.

### 4.2 Results from simulations

Figure 9 provides an overview of the typical nighttime emission and density profiles of the different relevant species. Only profiles close to local midnight were considered for the calculation of the mean curves. The climatological variations with

respect to local time and day of year for relative intensity of FeO and OFeOH are shown in Fig. 10. For the four $HO_2$-related emission processes listed in Table 3, the corresponding climatologies are displayed in Fig. 11. The reference intensities $\langle I \rangle$ for these climatologies are provided in Table 4. They are compared to the corresponding results for the X-shooter-based analysis, which involves the measurement of individual continuum features and the derivation of continuum components. For the intensity, the table also shows the correlation coefficients $r$ for the correlation of the model climatologies of the different

analysed emission processes with the variability patterns of the measured continuum features at 595 nm (f06a) and 1,510 nm (f15a) that are displayed in Fig. 4. The table also lists average centroid emission heights $\langle h_{\text{cen}} \rangle$ from the model-based nighttime climatologies, compared with the ranges indicated from the Q2DW-related analysis of the two continuum features in Sect. 3.5. Finally, the climatology-based effective solar cycle effect $\langle \text{SCE} \rangle$ is provided for the modelled and measured intensities.

### 4.2.1 FeO and OFeOH emission

The best-known structure of the nightglow continuum is the peak at about 595 nm, which is identified to be caused by FeO emission (Evans et al., 2010; Saran et al., 2011; Gattinger et al., 2011a; Unterguggenberger et al., 2017). The related WACCM climatology in Fig. 10a shows a primary maximum in May and a secondary one in October, whereas the lowest nocturnal averages occur around the turn of the year. The climatology for the 595 nm feature in Fig. 4a shows a similar seasonal variability



**Figure 9.** WACCM-related mean profiles of (a) volume emission rates of excited Fe-containing molecules (see Table 2), (b) number densities of Fe-containing molecules, (c) volume emission rates of excited $HO_2$ produced by different processes (see Table 3), and (d) number densities of species relevant for the production of excited $HO_2$ for local midnight (i.e. 8,760 profiles with local times of 23:48 and 00:18) at 24° S and 70° W.

pattern with only slight shifts in the peak positions. Although the moderate nocturnal decrease between austral autumn and spring in the WACCM data is not present in the measured climatology, the overall agreement is nevertheless satisfactory. The correlation coefficient $r$ for the grid cells with significant nighttime contribution is $+0.81$. Table 4 also shows an average centroid height for the FeO emission of Reaction R2 of $87.9\,\mathrm{km}$ (see also Fig. 9a), which is located between the low variability-based and maximum centroid emission heights from the analysis of Q2DW-related variations of the $595\,\mathrm{nm}$ variations in



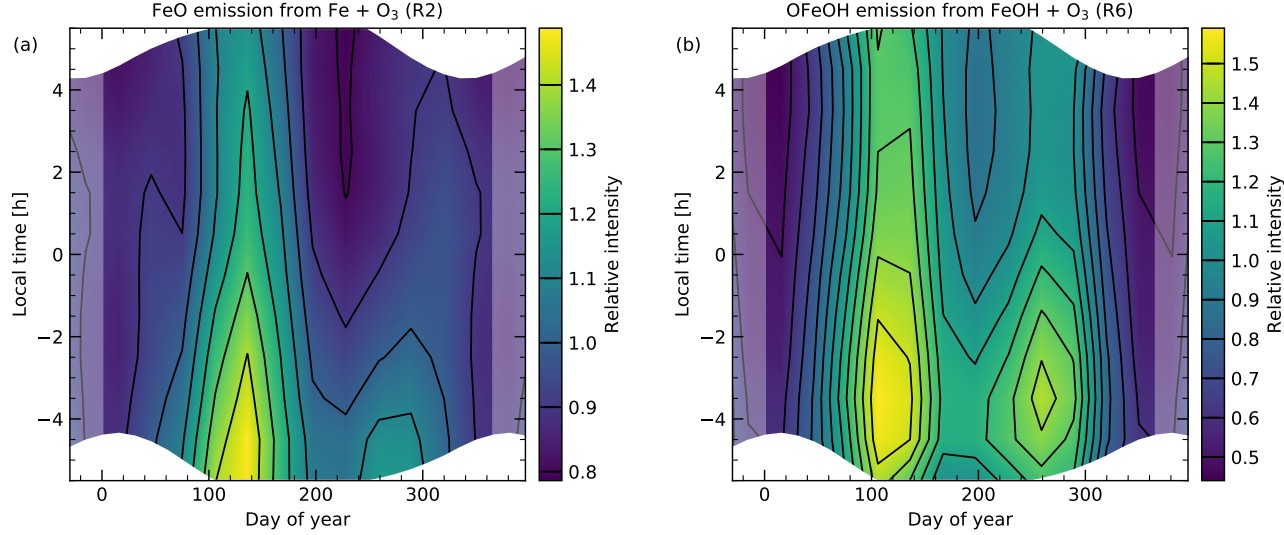

**Figure 10.** Climatologies of intensity relative to the mean for the vertically-integrated WACCM-simulated emission of (a) FeO and (b) OFeOH (see Table 2) at $24°$ S and $70°$ W. The climatologies were derived in a similar way as those in Fig. 4. With 92,064 selected nighttime data points, the subsample size for the relevant grid points was well above the limit of 200 (and even 400) used for the X-shooter data.

Sect. 3.5. Even the minimum and maximum values of the simulated climatology (which shows an increase with increasing
LT) of 86.3 and 88.9 km are inside this interval. Moreover, the moderate positive solar cycle effects from the modelled and measured climatologies of about $+16$ and $+11\%$ per 100 sfu agree quite well within their uncertainties (Sects. 4.1 and 3.4, respectively).

    Together with the spectral structure of the continuum in the VIS arm discussed in Sects. 3.1 and 3.2, this appears to be a robust identification of FeO chemiluminescence. However, the 595 nm feature is only a very small part of the entire well-
correlated FeO(VIS) component plotted in Fig. 3. Table 4 lists a mean intensity of the measured feature of about 27 R, whereas the FeO(VIS) component could emit about 2.9 kR. In contrast, the simulated mean intensity is only 170 R assuming a quantum yield (QY) of unity. This value is sufficient for the main peak of the orange bands, which would imply a QY of about 16%. This percentage is consistent with the result of Unterguggenberger et al. (2017) of $13 \pm 3\%$, which is also based on X-shooter measurements and WACCM simulations, although the samples and analysis approaches differ. If other features and more
underlying continuum is added to the calculation of the intensity, the simulated FeO emission budget is consumed quite fast. Adding just the continuum below the integration range of the 595 nm feature between 584 and 607 nm leads to 148 R for the mean continuum and 123 R if only the FeO(VIS) component is considered. Integration of the nightglow spectrum between 560 and 620 nm and subtraction of a constant flux that was measured at 500 nm returned 221 R. This kind of measurement was already performed by Saran et al. (2011) for ESI spectra taken at Mauna Kea (see Sect. 1). They found intensities up to 157 R
in two nights. A simulation based on FeMOD (Gardner et al., 2005) returned 61 R, which was then compared to a measured intensity of about 80 R. Interestingly, their ratio of 1.3 for measurement vs. model is the same that we obtain for our X-shooter





**Figure 11.** Climatologies of intensity relative to the mean for the vertically-integrated WACCM-simulated emission of excited $HO_2$ produced by three different processes (see Table 3) and the sum of them at $24°$ S and $70°$ W. Sample and calculation of the climatologies were the same as for Fig. 10. As the climatology in (a) shows a very large dynamical range, the colour scale was cut at a relative intensity of 4, which decreased the visualised range by more than a factor of 2.

and WACCM data. Moreover, Unterguggenberger et al. (2017) stated that the FeO main peak only contributes about 3.9% to the entire spectrum of the orange bands calculated by Gattinger et al. (2011a). For our case, this would be a total emission of 692 R. Unterguggenberger et al. (2017) also measured a fraction of $3.3 \pm 0.8\%$ of the main peak contribution to the emission between 500 and 720 nm. This is consistent with our results. We obtain 2.8% for the full nightglow continuum and 3.1% for the FeO(VIS) component.



**Table 4.** Comparison of nightglow continuum emissions from X-shooter spectra and WACCM simulations of potential emission processes

| Emission[a] | $\langle I \rangle$[b] (kR) | $r_{\text{f06a}}$[c] | $r_{\text{f15a}}$[d] | $\langle h_{\text{cen}} \rangle$[e] (km) | $\langle \text{SCE} \rangle$[f] ($10^{-2}$ sfu$^{-1}$) |
|---|---|---|---|---|---|
| 595 nm (f06a) | 0.027 | +1.000 | −0.305 | 85.2–89.2 | +0.107 |
| FeO(VIS) | 2.90 | +0.926 | −0.600 | | |
| FeO (R2) | 0.170 | +0.807 | +0.116 | 87.9 | +0.158 |
| OFeOH (R6) | 0.220 | +0.867 | −0.190 | 82.3 | +0.053 |
| FeO$_2$ (R10) | 0.0002 | +0.744 | +0.262 | 85.8 | +0.102 |
| 1,510 nm (f15a) | 1.37 | −0.305 | +1.000 | 80.0–84.1 | +0.042 |
| X(NIR) | 11.81 | −0.320 | +1.000 | | |
| HO$_2$ (R8) | 12.74 | +0.118 | +0.644 | 78.0 | +0.083 |
| HO$_2$ (R9) | 81.52 | +0.008 | +0.852 | 80.8 | +0.084 |
| HO$_2$ (R14) | 7.27 | +0.371 | +0.635 | 86.0 | +0.115 |
| HO$_2$ (sum) | 101.53 | +0.062 | +0.805 | 80.8 | +0.087 |

[a] continuum feature or component (X-shooter) or emission of given molecule and reaction in Tables 2 and 3 (WACCM)

[b] mean intensity from nighttime climatologies in Figs. 4, 10, and 11, or continuum component in Fig. 3

[c] correlation coefficient for correlation with climatology of 595 nm feature in Fig. 4a

[d] correlation coefficient for correlation with climatology of 1,510 nm feature in Fig. 4b

[e] range of possible centroid emission heights from Q2DW-related analysis in Sect. 3.5 (X-shooter) and mean centroid emission heights from nighttime climatologies (WACCM)

[f] mean relative solar cycle effect for the intensity for a change of the solar radio flux averaged for 27 days by 100 sfu from the corresponding nighttime climatologies (plotted in Fig. 6 for the X-shooter-related features)

Hence, the emissions of the apparent structures of the FeO orange bands in the nightglow continuum are already an issue for the model. Furthermore, the simulated intensity is more than an order of magnitude too low if the entire FeO(VIS) component spectrum is compared. This result is difficult to explain as WACCM agrees well with Fe concentrations measured by lidars
(Feng et al., 2013). Moreover, the rate coefficient for Reaction R2 has been measured in the laboratory (Helmer and Plane, 1994) and is close to the capture rate, i.e. the physical upper limit. A possible alternative production pathway of FeO would be the reaction of relatively abundant FeOH (see Fig. 9b) and H. However, this reaction is exothermic by only 61 kJ mol$^{-1}$, which is not sufficient to produce the observed spectrum. In principle, the latter would be possible by the sufficiently exothermic Reaction R10 in Table 2 that produces excited FeO$_2$. However, our simulation indicates that this emission is very faint (Fig. 9a).
The mean intensity in Table 4 is only 0.2 R.





**Figure 12.** Climatologies of WACCM-simulated relative number density at 80 km (left) and 85 km (right) for the chemically important, strongly variable species $O_3$ (top) and H (bottom) at $24°$ S and $70°$ W. Sample and calculation of the climatologies were the same as for Fig. 10.

The last imaginable Fe-related emission process would be related to Reaction R6 of FeOH and $O_3$. The resulting OFeOH radiation could cover the entire wavelength regime of the FeO(VIS) component, although wavelengths above 500 nm would be more likely (Sect. 4.1). As listed by Table 4, this nightglow process could produce up to 220 R, which is more than in the case of FeO. In order to be relevant for the FeO(VIS) component, the variability needs to be similar to the climatology of the 595 nm feature in Fig. 4a. Indeed, the pattern in Fig. 10b is very similar. The correlation coefficient is $+0.87$, which is even higher that in the case of the simulated FeO emission. Compared to the latter, the OFeOH climatology shows a more





prominent secondary peak in austral spring and a later nocturnal maximum. However, the discrepancies appear to be small enough that both emissions could contribute to the same NMF component. A clear difference are the emission profiles as shown in Fig. 9a. The mean nighttime centroid emission height for OFeOH is 82.3 km, which is 5.6 km lower than in the

case of FeO. It is also clearly lower than the derived range for the 595 nm feature. This is not an issue if this feature is mostly produced by the FeO-related Reaction R2 and the contribution of Reaction R6 between 564 and 680 nm is rather small in general. The latter constraint results from the OSIRIS-based emission profile with a peak at about 87 km (Sect. 3.5) that was derived for this wavelength range by Evans et al. (2010). The effective solar cycle effect as shown in Table 4 should also mainly be determined by Reaction R2. As the value for Reaction R6 is also not far away from the measurement for the 595 nm feature,

the contribution of both emission processes cannot be distinguished as in the case of the centroid heights.

These results suggests that OFeOH emission could significantly contribute to the FeO(VIS) component, although the emission spectrum (if any) is unknown, and the rate coefficient has not been measured. Here, it is set to the collision frequency, and a QY of unity is assumed to provide an upper limit to the contribution of this reaction. Despite this, the summed emission of Reactions R2 and R6 is still too low to explain the full nightglow continuum in the X-shooter VIS arm. So far, we have not

succeeded to identify another metal-related or any reaction that could explain the missing emission. In any case, it would be essential that the climatological variability pattern is mainly determined by the variations of $O_3$, which is a reactant in both Fe-related reactions. As shown by Fig. 12b for an altitude of 85 km, the semiannual pattern with maxima near the equinoxes and the main minimum in austral summer is a clear indicator of $O_3$ density changes in the mesopause region above Cerro Paranal.

### 4.2.2   HO$_2$ emission

The climatologies of the Fe-related emissions in Fig. 10 are very different from the variations of the 1,510 nm feature in Fig. 4b. The correlation coefficients are close to 0 (Table 4). Hence, we can focus our discussion of the possible emitter of the peak at 1,510 nm and the other structures of the X(NIR) continuum on HO$_2$, which appears to be the primary candidate for these emission features according to the discussion in Sect. 3.2.

The mean intensities in Table 4 show that there should be sufficient emission to produce the entire X(NIR) continuum. For the sum of the three reactions of about 102 kR, an effective QY of only about 12% would be necessary if we neglect possible emission beyond 1,800 nm (Sect. 3.3). If we only consider Reaction R9, it would be 14%. Although the individual QYs could be quite different, the reaction involving H and $O_2$ appears to be the dominating pathway. The intensity discrepancy is fairly large (see also Fig. 9c).

The climatology of Reaction R9 also indicates the best correlation with the climatology of the 1,510 nm feature in Fig. 4b. The corresponding $r$ value is about $+0.85$, whereas these coefficients are close to $+0.64$ for the two other reactions. The reasons for these differences are illustrated in Fig. 11. All reactions show a decrease of the intensity in the course of the night. However, the relatively constant rate of this decrease for (b) and (c) provides a much better agreement with the climatology of the 1,510 nm feature than the steep exponential drop of the intensity from Reaction R8 at the beginning of the night in (a). The



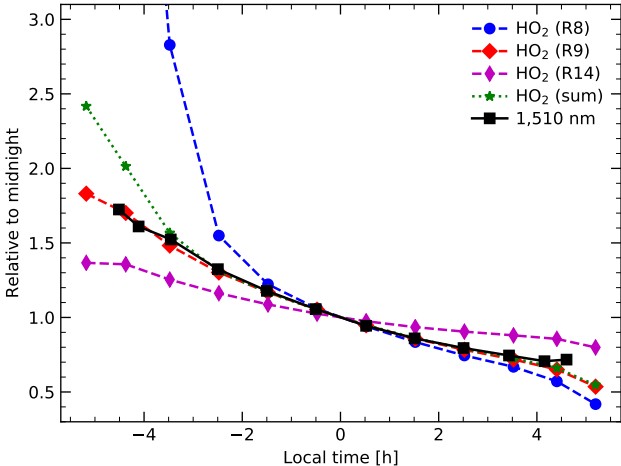

**Figure 13.** Climatology-based mean nighttime trend of WACCM $HO_2$ intensities for the individual reactions in Table 3 and the sum of them in comparison to the result for the climatology of the 1,510 nm emission plotted in Fig. 4b. All curves are provided relative to the mean intensity of the two data points close to midnight. The highest value for the curve for Reaction R8 (see legend), which is partly outside the plotted range, is 10.0. The given local times are the averages of the data sets that were used to calculate the climatological grid. The shorter time coverage of the curve for the 1,510 nm feature reflects the lack of observations with central LTs close to twilight.

latter is related to the decay of the $O_2(a^1\Delta_g)$ population produced by $O_3$ photolysis at daytime. At later LTs, $O_2(a^1\Delta_g)$ is then mainly produced by Reaction R15 in our model.

     For an easier comparison, Fig. 13 shows the mean nocturnal trends from the different climatologies scaled to midnight. Reaction R8 is a clear outlier. Moreover, the intensity from Reaction R14 appears to decrease too slowly compared to the curve for the 1,510 nm emission. On the other hand, Reaction R9 seems to match almost perfectly. Of course, this could

also be coincidence to some extent. A check of the monthly nocturnal trends also indicates clear deviations. The trend for the simulated emission tends to be steeper from November to March and flatter from April to October. Although possible systematic deviations of the model are difficult to estimate, this result favours Reaction R9. The large contribution of the latter to the summed intensity means that the nocturnal trends of the summed intensity and due just to Reaction R9 look very similar in Fig. 13. The only noteworthy discrepancy occurs at the beginning of the night due to the high intensity related to Reaction R8

(see also Fig. 11d). However, the difference is still relatively small at the earliest data point for the 1,510 nm emission. Hence, some contribution of this reaction to the total emission cannot be excluded, but it should not be significantly higher than modelled for equal QYs. This statement refers to the branch of the $HO_2$ production by Reaction R8 related to daytime $O_3$ photolysis. The mean intensity after midnight is only 5.0 kR, i.e. 39% of the mean of the full nighttime climatology. However, this value is quite uncertain as our discussion of the nocturnal $O_2(a^1\Delta_g)$ generation in Sect. 4.1 illustrates.

In order to obtain a rough estimate of the quality of our assumptions, we made a simple conversion of the $O_2(a^1\Delta_g)$ densities from WACCM (mean midnight profile in Fig. 9d) to emission rates assuming a QY of unity and using an effective Einstein-A coefficient of $2.28 \times 10^{-4}\,\mathrm{s}^{-1}$ for the $O_2$(a-X)(0-0) band emission at 1,270 nm (Noll et al., 2016). We then compared the



vertically integrated emission rates with those from SABER measurements in the corresponding channel. For this purpose, we used the 4,496 profiles that were collected for Cerro Paranal by Noll et al. (2017). Next, we performed a similar analysis

of the nocturnal intensity trend as described by Noll et al. (2016), i.e. we fitted an exponential function and a constant for the time after sunset. As the WACCM data set is quite large, we performed this for each month separately and averaged the results. The fit functions were scaled to the mean of the second half of the night. For the intensities for the entire column above 40 km, these reference constants are 29.7 kR for WACCM and 102.0 kR for SABER. However, the vertical emission distribution for $O_2$(a-X)(0-0) with a centroid emission height of about 89 km around midnight (Noll et al., 2016) is quite

different from the WACCM-based profile shown in Fig. 9d. Hence, we restricted the comparison to the height range between 80 and 85 km. Then, we obtain reference intensities of 18.4 kR (62%) for WACCM and 15.4 kR (15%) for SABER. For the exponential component, we fitted time constants of about 74 and 76 min, which are very close to the radiative lifetime of about 73 min from the Einstein-A coefficient. For the entire vertical column, the lifetimes decrease to 55 and 50 min in agreement with the results from Noll et al. (2016). The change is related to the higher impact of collisional deactivation at lower altitudes.

In any case, the good agreement of the WACCM and SABER time constants indicate that the related temporal variations are well simulated by WACCM. For the integration from 80 to 85 km, the intensities of the exponential component are 71.8 kR for WACCM and 169.9 kR for SABER 60 min after sunset. As these values are 21 to 22% of the corresponding intensities for the whole column, WACCM also performs quite well with respect to the vertical distribution of $O_2$(a-X)(0-0) produced by $O_3$ photolysis. However, the WACCM-related intensity is clearly lower. On the other hand, the corresponding intensity for

the nighttime production is somewhat higher than the value from SABER. If we assume a constant ratio of the WACCM and SABER intensities for the entire night, then the branching ratio in Reaction R15 needs to be lowered to about 33%. Of course, the uncertainties of this comparison are quite high. For example, the height-dependent impact of collisions can play a role. However, the results appear to show that a distinctly higher contribution of Reaction R8 to the generation of excited $HO_2$ in the night than simulated is unlikely, which strengthens the importance of Reaction R9.

The climatology of the 1,510 nm feature in Fig. 4b indicates semiannual variations with a main maximum in austral summer and a secondary maximum in winter. The $HO_2$-related climatologies in Fig. 11 show discrepancies with respect to the seasonal variations. Nevertheless, the highest intensities are also present in summer for Reaction R9. If the first few hours of the night are excluded, this also appears to be true for Reaction R8. The climatologies of these reactions are also characterised by minimum intensities in May to June slightly depending on LT. The seasonal pattern of Reaction R14 is very different as it indicates

a semiannual oscillation with maxima around the equinoxes, which explains the low correlation coefficient in Table 4. This pattern is obviously related to the important role of $O_3$ in the production of $HO_2$. The similarities are clearly visible in the top row of Fig. 12, where the climatologies of the $O_3$ densities at 80 and 85 km are displayed. Moreover, the bottom row of the same figure, which shows H for the same heights, explains the location of the seasonal maxima and minima for the other reactions. This makes sense as H is the only strongly variable reactant in the production of $HO_2$ via Reaction R9.

As Reaction R8 requires the previous generation of $HO_2$, there is also an impact of the H variability on this reaction. It also explains why the climatology of Reaction R14 still better correlates with the variations of the 1,510 nm feature than the 595 nm





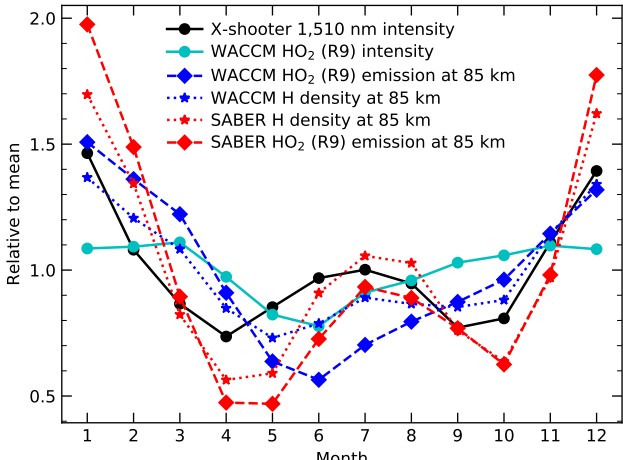

**Figure 14.** Seasonal variations of monthly averages of time series of the X-shooter-based 1,510 nm intensity (7,931 30 min bins), the WACCM $HO_2$ intensity for the dominating Reaction R9 (92,064 time steps), the corresponding $HO_2$ emission rate and H density at 85 km, and the SABER-based $HO_2$ emission rate and H density at the same altitude (16,079 profiles) for Cerro Paranal. The different curves (see legend) are provided relative to the mean for the 12 months.

feature (Table 4). Hence, this pathway (despite some similarities with the Fe-related emissions discussed in Sect. 4.2.1) does not appear to be a suitable contributor to the FeO(VIS) continuum component.

In order to also identify H as the main driver of the climatological variations of the 1,510 nm feature, we need to find the reason for the missing secondary maximum in the WACCM simulation in winter. Mlynczak et al. (2014) performed global H retrievals based on the 2.1 μm channel of SABER (Russell et al., 1999). As a result, they found semiannual seasonal variations at 84 km and a latitude of 16.5° S with a secondary maximum in austral summer. This pattern was confirmed by a comparison with older results at 20° S from Thomas (1990) based on measurements with the near-IR spectrometer on the Solar Mesosphere Explorer. Mlynczak et al. (2018) improved the OH-based retrieval algorithm for O and H, which reduced the densities. We

already used this data set for the years 2002 to 2014 for an analysis of K emissions above Cerro Paranal (Noll et al., 2019). This allows us to reuse these data for the study of the seasonal H variations. For this purpose, we calculated simple monthly mean values for the SABER and WACCM time series. For an altitude of 85 km, Fig. 14 confirms that there is a weak secondary maximum in the SABER H density. Moreover, the primary peak in summer is more pronounced than in the WACCM simulation. In the next step, we compared $HO_2$ emission rates. For the dominating Reaction R9, these were calculated from

the SABER densities of H and air and the rate coefficients in Table 3. Only the weakly-varying $O_2$ volume mixing ratio was taken from WACCM. The results for 85 km are also presented in Fig. 14. The deviations from the variations of H are small and similar for both data sets, which illustrates the dominating role of H for the $HO_2$ emission variability. The comparison of the WACCM $HO_2$ emission at 85 km and for the entire vertical column indicates a smoothing of the seasonal pattern in the latter case. We cannot test the same for the SABER data as the noise in the H retrievals quickly increases at lower altitudes due to the

weaker emission of the OH(8-6) and OH(9-7) bands that were essentially used for the derivation of the H density (Mlynczak





et al., 2018). This is also the main reason why the decreasing nocturnal trend in the $HO_2$ emission could not be studied in this way. Figure 12 indicates that this trend is obviously generated distinctly lower than 85 km. Nevertheless, if we assume that the smoothing in the seasonal pattern in the WACCM data is similar for the observations, it is likely that we would obtain a variability structure that resembles the curve for the 1,510 nm feature plotted in Fig. 14. Hence, the differences between modelled

and observed emission variations appear to be mostly caused by the WACCM-based reproduction of temporal changes in the H density.

Table 4 shows the mean centroid emission altitudes from the corresponding climatologies of the different production processes of $HO_2$ emission. Only the height of about 81 km for Reaction R9 is in the range from about 80 to 84 km for the average profile derived from the X-shooter-based analysis in Sect. 3.5. The agreement looks even more promising if it is considered that

O, which is crucial for the upper limit of 84 km, is not involved in the chemical production of excited $HO_2$ for this pathway. The simulated climatological variations show increasing heights during the night and the highest values in austral summer with maximum deviations from the mean lower than 2 km in most cases. The emission of Reaction R8 just reaches 80 km close to sunrise, whereas the centroid heights are even below 70 km at the beginning of the night due to the high concentrations of $O_2(a^1\Delta_g)$ related to daytime $O_3$ photolysis in the lower mesosphere (Noll et al., 2016). Reaction R14 shows a similar vari-

ability pattern as Reaction R9 but with a mean value of 86 km the emission is too high. The reason is the impact of $O_3$, which shows its density peak at a higher altitude than H in the mesopause region in Fig. 9d. This plot also indicates that the density distribution of all $HO_2$ peaks at 78 km, which is lower than the emission maximum of excited $HO_2$ shown in Fig. 9c.

Finally, Table 4 shows that at least the results for Reactions R8 and R9 appear to agree with a positive but weak solar cycle effect that was derived for the 1,510 nm feature in Sect. 3.4. Although the uncertainties are of the order of several per cent, the

almost doubled deviation from the measured value compared to the other reactions, make Reaction R14 less likely with respect to the response to solar activity.

Our investigation of the three proposed production mechanisms for excited $HO_2$ shows promising results. It is likely that $HO_2$ is the radiating molecule X that produces the 1,510 nm feature and the strongly correlated X(NIR) continuum. Moreover, the recombination reaction of H and $O_2$ with participation of an additional collision partner is the most probable production

process for excited $HO_2$. All investigated properties such as total emission, nocturnal and seasonal variations, emission heights, and solar cycle effect point to this interpretation. It also helps that chemiluminescence by this mechanism was already observed in the laboratory between about 800 and 1,550 nm (Holstein et al., 1983), although an extension to higher wavelengths would be important to test the presence of the 1,620 nm feature (Fig. 3). Reaction R8 that involves $O_2(a^1\Delta_g)$ can only be a minor emission source, otherwise the observed nocturnal trend would not fit. It is also more challenging to produce emission below

1,270 nm for this process. Finally, Reaction R14 that involves $O_3$ shows clear discrepancies in the emission properties which can only be tolerated if the contribution to the total emission is very small.





## 5   Conclusions

Our analysis of the nightglow (pseudo-)continuum with high-quality X-shooter data essentially reveals two contributions in the wavelength range between 300 and 1,800 nm if remnants from different $O_2$ bands are excluded.

Our results of the correlation analysis of continuum structures and non-negative matrix factorisation (NMF) of the continuum variability show that the peak at 595 nm is well correlated with other features and the underlying continuum in a wide wavelength range, especially between about 500 and 900 nm. The variations as mainly studied for the feature at 595 nm reveal a climatology with a mixture of semiannual and annual oscillation with a main maximum in April/May and a main minimum in January that confirms previous results based on a smaller sample. For the first time, we estimated the effective solar cycle effect

and found a weakly positive relation. Using an approach for the estimate of effective emission heights based on the analysis of a strong passing Q2DW that was initially developed for OH lines, we obtained a range for the mean centroid height between about 85 and 89 km.

In previous studies, the feature at 595 nm was identified as the main peak of the FeO orange bands. Own simulations of the chemiluminescence from the reaction of Fe and $O_3$ with WACCM can reproduce most of the measured properties of the

emission, which suggests that the NMF component dominating the X-shooter VIS arm could have contributions from various FeO bands. However, WACCM returns a maximum mean intensity of only 170 R, whereas the whole correlated spectrum could have 2.9 kR. We discovered that potential OFeOH emission (with unknown spectral distribution) from the reaction between FeOH and $O_3$ would have a very similar climatology according to our simulations. Nevertheless, this reaction would only add up to 220 R. Therefore, a major discrepancy remains. If there is another emitter, the basic precondition for a good correlation

with the 595 nm feature appears to be that the variability is mainly determined by $O_3$.

The second continuum component dominates the X-shooter NIR arm. In particular, a strong narrow peak at about 1,510 nm and a secondary feature at about 1,620 nm were found, which indicates a complex band system. Our best estimate of the average intensity of the entire system is about 12 kR. The seasonal variations with maxima near the solstices are actually in opposition to those of the 595 nm feature. There is also a clear decrease of the intensity in the course of the night for the entire

year. The solar cycle effect is only weakly positive and the average effective emission height appears to be most likely between about 80 and 84 km for the 1,510 nm feature.

The most promising candidate for the emitter is $HO_2$. Existing near-IR spectra from the laboratory suggest that the 1,510 nm feature could be the vibrational (200-000) transition of the electronic ground state $^2A''$. There would also be an explanation of the enhanced emission near 1,270 nm, where only a part appears to be due to residual $O_2$ emission. Other features from

the experiments could not be checked due to gaps in our continuum spectrum. We investigated different potential production processes of excited $HO_2$ with WACCM. The recombination reaction between H and $O_2$ under participation of another collision partner showed the best performance. It is the main production process of $HO_2$ in the mesosphere. With a modelled maximum mean radiance of 82 kR, a moderate quantum yield of the reaction would be sufficient to produce the continuum in the X-shooter NIR arm. Moreover, this process indicates the best agreement with respect to the climatological variations. Re-

maining discrepancies (especially a missing secondary peak in austral winter) can be explained by deviations of the modelled



H densities from those of SABER-based retrievals. The simulated weak solar cycle effect and the average centroid emission height of about 81 km also show good agreement. Finally, the observed chemiluminescence in the laboratory for this mechanism indicated emission down to about 800 nm, which is consistent with the shape of the derived continuum component. As wavelengths above about 1,550 nm were not studied in the only known laboratory experiment, there is no evidence of the existence of the 1,620 nm feature, so far. The other studied potential emission processes appear to be much less efficient. Relatively weak or even negligible emission rates are probably related to the direct radiative recombination of H and $O_2$, the reaction of H and $O_3$, and collisions of $HO_2$ with $O_2(a^1\Delta_g)$. The latter would produce a steep decline of the emission after dusk due to the decay of the population of excited $O_2$ molecules produced by $O_3$ photolysis, which is not observed in the X-shooter data. The reaction involving H and $O_3$ would generate a very different seasonal variability as observed.

The intriguing discoveries of this study will certainly trigger further investigations for a better understanding of the chemistry and dynamics in the Earth's mesopause region. The origin of the whole VIS-arm continuum still needs to be solved. The study also revealed that the nighttime production of $O_2(a^1\Delta_g)$ is not understood well, although these excited molecules are essential for the strong emission at 1,270 nm. These examples illustrate that there are still many things at these altitudes that we do not know.

*Data availability.* The basic X-shooter data for this project originate from the ESO Science Archive Facility at http://archive.eso.org and are related to various observing programmes that were carried out between October 2009 and September 2019. The raw spectra were processed (using the corresponding calibration data) and then analysed. The NIR-arm data were already used for the study of OH emission lines (Noll et al., 2022a, 2023b). Some results of these investigations, which are available via the public repository Zenodo (Noll et al., 2022b, 2023c), were also considered for this study. We performed dedicated WACCM6 runs with modified chemistry for the years from 2003 to 2014. The crucial results are stored at the University of Leeds. We also made use of TIMED/SABER data sets that were already collected for previous studies for Cerro Paranal from the SABER website at http://saber.gats-inc.com. These are the v2.0 products from 2002 to 2015 analysed by Noll et al. (2017) and the improved O and H retrievals described by Mlynczak et al. (2018) for the years 2002 to 2014 that were used by Noll et al. (2019). Results from the study of v2.0 products from January and February 2017 by Noll et al. (2022a) were also considered. A comprehensive collection of data of our analysis (especially with respect to the plotted data) is provided by Zenodo at https://zenodo.org/record/8335836 (Noll et al., 2023a).

*Author contributions.* SN designed and organised the project, performed the preparation and analysis of the X-shooter spectra, visualised the results based on X-shooter, WACCM, and SABER data, and is the main author of the paper text. The co-authors contributed to the improvement of the paper content. Moreover, JP designed the WACCM runs, investigated the involved chemistry, and significantly influenced the scientific discussion. WF performed the WACCM simulations and a preliminary analysis of the data. KK significantly contributed to the discussion of the chemistry. In particular, he first proposed $HO_2$ as possible emitter. WK carried out the basic processing of the X-shooter spectra. CS contributed to the discussion of the X-shooter-based analysis. MB was involved in the management of the project. SK managed the infrastructure for the processing and storage of the X-shooter data and contributed to the discussion of the measured continuum features.



*Competing interests.* Co-author John Plane is a member of the editorial board of Atmospheric Chemistry and Physics. The authors do not have other competing interests to declare.

1050    *Acknowledgements.* Stefan Noll was financed by the project NO 1328/1-3 of the German Research Foundation (DFG). John Plane and Wuhu Feng were supported by grant NE/T006749/1 from the UK Natural Environment Research Council. We thank Sabine Möhler from ESO for her support with respect to the X-shooter calibration data.



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
