# Peer review of "Structure, variability, and origin of the low-latitude nightglow continuum between 300 and 1,800 $\rm nm$ : Evidence for $\rm HO_2$ emission in the near-infrared"

_EGUsphere, 2023_

## Author Response (AR1)

**Response to 'Comment on egusphere-2023-2087' by Anonymous Referee #1**

*... In summary, this is an outstanding study that in addressing the identity of the nightglow continuum raises new and important questions in atmospheric chemistry that grant additional laboratory, field and modelling work. Therefore I strongly support the publication of this manuscript in ACP."*

We are very pleased that Reviewer #1 acknowledges our manuscript on the structure, variability, and origin of the nightglow continuum as "an outstanding study" which "raises new and important questions in atmospheric chemistry".

*"I have found a small typo in panel a of Figure 9: FeO3 should be FeO."*

We thank the reviewer for spotting the typo in Fig. 9. We have corrected the plot.

**Response to 'Comment on egusphere-2023-2087' by Anonymous Referee #2**

*"... The paper is very well written, providing a comprehensive overview of the state of the art and describing a very well designed and careful analysis of the dataset in great detail, from calibration and corrections of the raw spectra to the decomposition of the continuum and the interpretation of the observed modes of variability with WACCM model results. As the NMF method was new to me, I'd have liked a bit more information about this. I also have a few other minor points listed below."*

We thank Reviewer #2 for the valuable comments. We are pleased about the positive feedback with respect to the quality of our study.

*"Line 105-106, low resolving power of what, the FORS 1 spectrograph? Please clarify in the text."*

Yes, this assumption is correct. We have added "of FORS 1" after "low resolving power".

*"Caption of Fig. 1: "the original sky spectrum is marked by the cyan", "after subtraction of these componentns, the cyan spectrum marks the nightglow emission": please clarify whether the cyan is the original sky spectrum, or the nightglow emissions – my understanding here is that these are not the same, but that the nightglow emission is the original sky spectrum minus the modelled background contributions marked by the green line"*

We are sorry that the description appears to be confusing. The original sky spectrum corresponds to the difference between the cyan and the green curve, whereas the observed nightglow spectrum corresponds to the difference between the cyan curve and the zero line. We have revised the text in the following way:

"The green curve indicates the summed modelled contributions by scattered moonlight (not relevant here), zodiacal light, scattered starlight, and thermal emission of the telescope and instrument (multiplied by -1 for the plot) that were subtracted from the original sky spectrum. The cyan spectrum (or the overplotted red curve in the case of complete overlap) limited by the dotted zero line is the result of this subtraction and marks the nightglow emission.".

*"Line 216: does absorption by $NO_2$ play a role in the visible?"*

Indeed, $NO_2$ absorbs in the visible, especially at blue wavelengths. However, combining the wavelength-dependent absorption cross section (e.g., Burrows et al., 1998, DOI:10.1016/S0022-4073(97)00197-0) and typical vertical column depths for clean environments dominated by the stratospheric contributions (e.g., Leue et al., 2001, DOI:10.1029/2000JD900572) indicates optical depths clearly lower than 0.01 at zenith. Hence, the impact of $NO_2$ can be neglected for the typical zenith angles of astronomical observations. A typical calculated absorption spectrum for Cerro Paranal is shown in Fig. 1 of Smette et al. (2015, DOI:10.1051/0004-6361/201423932). We now provide this paper as a literature reference after the list of relevant absorbing molecules.

*"Lines 300 – 330: the NMF was new to me. I would probably have tried empirical orthogonal functions because I'm more familiar with that, and also for the added value that you don't have to make assumptions about the number of independent modes of variation; but that is probably a matter of taste as you apparently tested different values for the modes of variation. However, I am wondering about the need to imply weightings, which you yourself mention that their choice is somewhat arbitrary. So, how exactly are weightings imposed in the NMF, via covariance matrices? And how much does the choice of weightings affect the results? This should be explained more clearly. Maybe you could just include the equation you solved. Also, I found the statements "this choice had little impact on the structure of the solution" (lines 327-328) and "small changes in the scaling factors can lead to a very different regime of solutions" somewhat contradictory; maybe you can clarify that."*

We have already used NMF for the decomposition of the climatologies of 298 OH lines (Noll et al., 2023b). NMF shows good performance for non-negative additive quantities. Hence, it is appealing to apply it for the decomposition of the nightglow continuum. In principle, only the number of components needs to be defined, which is relatively easy for our case. However, the default run led to a lack of separation between the FeO(VIS) and faint $O_2$(UVB) components. Therefore, we introduced the wavelength-dependent weighting of the input spectra to increase the weight of the previously-identified reference features. Of course, this increased the number of parameters we had to adjust. For this reason, we defined a simple cost function in order to identify the most realistic solution by an automatic search. In the end, we obtained convincing and robust spectra for the two major components of the nightglow continuum. In order to improve the reader's understanding, we now discuss this approach by means of mathematical expressions. We have also revised the statements which the reviewer found confusing.

In the first paragraph of Sect. 3.2, we have extended the introduction of the basic NMF:

"Our approach was to use non-negative matrix factorisation (NMF; e.g., Lee and Seung, 1999; Noll et al., 2023b) as it is well suited for additive components without negative values. NMF approximately decomposes an $m$ x $n$ matrix $\mathbf{X}$ without negative elements into two non-negative matrices $\mathbf{A}$ and $\mathbf{B}$ with sizes $m$ x $L$ and $L$ x $n$, respectively, by usually minimising the squared Frobenius norm of $\mathbf{X}$ - $\mathbf{AB}$.".

In the second paragraph, we now show the mathematics related to our modified NMF approach:

"Consequently, the algorithm minimised $\|\mathbf{X'} - \mathbf{A'B}\|^2_{\text{Fro}}$, where $\mathbf{X'} = \mathbf{SX}$ and $\mathbf{A} = \mathbf{S}^{-1}\mathbf{A'}$ with $\mathbf{S}$ being a diagonal $m$ x $m$ matrix containing the wavelength-dependent scaling factors.".

In the same paragraph, we have extended the description of the construction of the cost function:

"To find the best scaling factors, we defined a cost function $C$ that uses the relative contributions $f_{ij}$ of the $L = 4$ component spectra to the four corresponding feature windows as defined above, i.e. we attempted to minimise $C = 1 - \sum_{i=1}^{L} w_{i,\max} f_{i,\max}$, where $w_{i,\max}$ and $f_{i,\max}$ being the weight and relative flux of the most relevant feature window $j$ contributing to component $i$. Equal weights of 0.25 for the four feature windows favoured solutions with particularly large contributions of the two $O_2$-related components.".

At the end of the second paragraph, we have significantly revised the discussion of the performance of our cost function. We accept that the original text was confusing:

"This (somewhat arbitrary but non-critical) definition was sufficient to easily distinguish between solutions with good component separation (best $C$ of 0.26) and those where the separation failed, especially in the case of $O_2$(UVB) and FeO(VIS) (best $C$ of 0.32). The relation between the scaling factors in **S** and the structure of the component spectra turned out to be complex. However, the variations within a certain class of solutions tended to be relatively small. Hence, the solutions related to a satisfactory separation of the four components as indicated by low $C$ are relatively robust. In any case, there are two major components that dominate the visual and near-infrared ranges.".

Hence, the original statement on "a very different regime of solutions" only referred to the complex dependence of the solutions on the applied feature scaling parameters. This was the reason for the use of the SHGO search algorithm with many sampling points. However, the structure of the resulting component spectra looked very similar for low values of the cost function. Hence, the results can be considered as robust.

*"Lines 579-580 – the 2D climatologies for the two features look quite similar, with lower values in austral autumn, higher values in austral winter and spring – is it possible that there is an aliasing in the selection of data between time of year and solar activity?"*

Our X-shooter samples for both continuum features only show a relatively small dependence of the solar radio flux on local time and month. The climatological solar radio fluxes only vary between 89 and 109 sfu for nighttime bins (cf. discussion in Sect. 3.3). The structure of the binned 10-year data set is illustrated in Fig. 1 of Noll et al. (2023b). There is no obvious bias. Even if we assumed significant inhomogeneities, it would be unlikely to observe similar structures in the SCE climatologies as they are relative to the intensity climatologies, which are quite different for both continuum features.

In the context of the discussion of similarities, we have added the following sentence at the end of the paragraph: "Despite a low correlation coefficient of +0.31, the SCE climatologies of both features appear to be more similar than in the case of the intensity climatologies displayed in Fig. 4.".

We do not provide a more detailed discussion of these results as the interpretation appears to be quite complicated.

*"Section 3.5, Figure 7: even if we accept that only the phase is needed to here, the agreement between the X(1510nm) and the fitting model result is really not very good. In this sense, I think the conclusion here is that the results don't contradict an emission altitude of 80+- a few km, but they don't conclusively show that this really is the emission altitude. So, the conclusion for X(1510nm) seems to be that the emission altitude is not in contradiction to X being HO$_2$."*

As stated in the text, the relative rms of the fit of the X(1510nm) time series in Fig. 7 is only 14%. This percentage is very similar to the results for the two OH lines shown in Fig. 8 and much better than in the case of FeO(595nm) (23%). As the OH-based analysis by Noll et al. (2022a) resulted in realistic centroid emission altitudes (with estimated uncertainties lower than 1 km), we think that the phase derivation for X(1510nm) is also convincing. At least, it is relatively safe to conclude that the centroid height for X(1510nm) is lower than for any OH line. Nevertheless, it is challenging to provide an absolute height due to the uncertainties related to converting variability-based effective heights to mean centroid emission heights. Hence, we only conclude that the "the average effective emission height appears to be most likely between 80 and 84 km" in Sect. 5. In our opinion, this is a sufficiently cautious statement with respect to the uncertainties. In fact, it implies that "the emission altitude is not in contradiction to X being $HO_2$".

*"Line 725-726: in which altitude range was WACCM6 nudged against MERRA2?"*

After "(Molod et al., 2015)", we have added "below 50 km. Then, the amount of relaxation is linearly reduced. Above 60 km, the model is free running (cf. Marsh et al., 2013).".

Note that Marsh et al. (2013, DOI:10.1002/jgrd.50870) used the range from 40 to 50 km for the transition. However, our choice is consistent with the default specified dynamics version of WACCM.

*"Table 4, Figures 10 and 3: both from the visual impression comparing Fig 10 a) and b) to Figure 4 a) / 5 b), and from the correlation coefficients in Table 4, I would say that the variation of OFeOH fits slightly better than FeO; it seems to reproduce the very slow decline during night better. So I would tentatively conclude from this that FeOH + $O_3$ likely contributes to the observed feature, and also add this in the abstract (lines 15-16)."*

We agree that OFeOH is an appealing candidate with respect to the production of the FeO(VIS) continuum component, although we do not know the structure of the spectrum. We discuss the possible role of this molecule in the last two paragraphs of Sect. 4.2.1. It is important to distinguish between the FeO(VIS) component and the 595 nm feature. In the latter case, a significant contribution of OFeOH is relatively unlikely as the simulated altitude distribution of the emission does not match the observations, and the laboratory and theoretical spectra of FeO show good agreement.

As proposed, we have added an OFeOH-related sentence at the end of the abstract: "A possible (but nevertheless insufficient) process could be the production of excited OFeOH by the reaction of FeOH and $O_3$.".